# Training for Stable Explanation for Free

**Chao Chen**[1]  **Chenghua Guo**[2]  **Rufeng Chen**[3]  **Guixiang Ma**[4]
**Ming Zeng**[5]  **Xiangwen Liao**[6]  **Xi Zhang**[2]  **Sihong Xie**[3*]
[1]School of Computer Science and Technology, Harbin Institute of Technology (Shenzhen), China
[2]Key Laboratory of Trustworthy Distributed Computing and Service (MoE),
Beijing University of Posts and Telecommunications, China
[3]Artificial Intelligence Thrust, The Hong Kong University of Science and Technology (Guangzhou),
China [4]Intel, USA   [5]Carnegie Mellon University, USA
[6]College of Computer and Data Science, Fuzhou University, China
cha01nbox@gmail.com   {chenghuaguo,zhangx}@bupt.edu.cn
rchen514@connect.hkust-gz.edu.cn   jean.maguixiang@gmail.com
ming.zeng@sv.cmu.edu   liaoxw@fzu.edu.cn
*Corresponding to: xiesihong1@gmail.com

## Abstract

To foster trust in machine learning models, explanations must be faithful and stable for consistent insights. Existing relevant works rely on the $\ell_p$ distance for stability assessment, which diverges from human perception. Besides, existing adversarial training (AT) associated with intensive computations may lead to an arms race. To address these challenges, we introduce a novel metric to assess the stability of top-$k$ salient features. We introduce R2ET which trains for stable explanation by efficient and effective regularizer, and analyze R2ET by multi-objective optimization to prove numerical and statistical stability of explanations. Moreover, theoretical connections between R2ET and certified robustness justify R2ET's stability in all attacks. Extensive experiments across various data modalities and model architectures show that R2ET achieves superior stability against stealthy attacks, and generalizes effectively across different explanation methods. The code can be found at `https://github.com/ccha005/R2ET`.

## 1   Introduction

Deep neural networks have proven their strengths in many real-world applications, and their explainability is a fundamental requirement for humans' trust [25, 62]. Explanations usually attribute predictions to human-understandable basic elements, such as input features and patterns under network neurons [35]. Given the inherent limitations of human cognition [69], only the most relevant top-$k$ features are presented to the end-users [89]. Among many existing explanation methods, gradient-based methods [55, 24] are widely adopted due to their inexpensive computation and intuitive interpretation. However, the gradients can be significantly manipulated with neglected changes in the input [16, 23]. The susceptibility of explanations compromises their integrity as trustworthy evidence when explanations are legally mandated [25], such as credit risk assessments [4].

**Challenges.** As shown in Fig. 1, novel explanation methods [75, 77] and training methods [13, 17, 88] are proposed to promote the robustness of explanations. We focus on training methods since novel explanation methods require extra inference (explanation) time and may fail the sanity check [1]. Existing works [17, 88, 13, 16] study the explanation stability (robustness) through $\ell_p$ distance. However, as shown in the right part of Fig. 1, a perturbed explanation with a small $\ell_p$ distance to the original can exhibit notably different top salient features. Therefore, relying on $\ell_p$ distance as a metric for optimizing the explanation stability fails to attain desired robustness, indicating the necessity for

38th Conference on Neural Information Processing Systems (NeurIPS 2024).

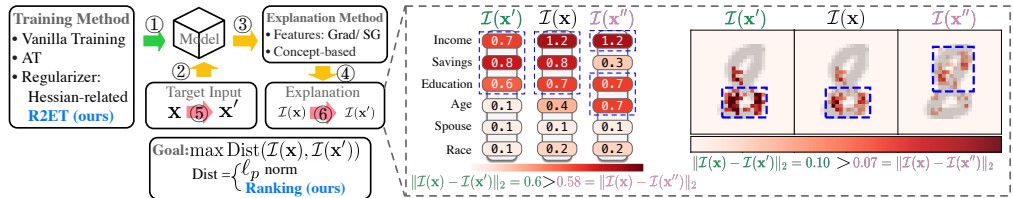

Figure 1: **Left**: *Green* (①): Model training. *Yellow* (②-④): Explanation generation for a target input. *Red* (⑤-⑥): Adversarial attacks against the explanation by manipulating the input. **Right**: Two examples of the saliency maps (explanations) show that smaller $\ell_p$ distances do not imply similar top salient features. $\mathcal{I}(\mathbf{x}'')$ has a smaller $\ell_2$ distance from the original explanation $\mathcal{I}(\mathbf{x})$, but manipulates the explanation more significantly (by top-$k$ metric) shown in blue dashed boxes. Statistically, $\mathcal{I}(\mathbf{x}')$ has a 67% top-3 overlap in the tabular case, and 36% top-50 overlap in the image, compared with $\mathcal{I}(\mathbf{x}')$'s 100% and 92% top-$k$ overlap, respectively.

a novel ranking-based distance metric. Second, unlike robustness of prediction and $\ell_p$ distance based explanation, analyses of robustness on ranking-based explanations are challenging due to multiple objectives (feature pairs), which are dependent since features are from the same model. Finally, adversarial training (AT) [86, 74] has been adopted for robust explanations, yet it potentially leads to an attack-defense arms race. AT conditions models on adversarial samples, which intrinsically rely on the objectives of the employed attacks. When defenders resist well against specific attacks, attackers will escalate attack intensity, which iteratively compels defenders to defend against fiercer attacks. Besides that, AT is time-intensive due to the search for adversarial samples per training iteration.

**Contributions.** We center our contributions around a novel metric called "ranking explanation thickness" that precisely measures the robustness of the top-$k$ salient features. Based on the bounds of thickness, we propose R2ET, a training method, for robust explanations by an effective and efficient regularizer. More importantly, R2ET comes with a theoretically certified guarantee of robustness to obstruct all attacks (Prop. 4.5), avoiding the arms race. Besides certified robustness, we theoretically prove that R2ET shares the same goal with AT but avoids intensive computation (Prop. 4.6), and the connection between R2ET with constrained optimization (Prop. 4.7). We also analyze the problem by multi-objective optimization to prove explanations' numerical and statistical stability in Sec. 5.

**Results.** We experimentally demonstrate that: *i*) Prior $\ell_p$ norm attack cannot manipulate explanations as effectively as ranking-based attacks. R2ET achieves more ranking robustness than state-of-the-art methods; *ii*) Strong evidence indicates a substantial correlation between explanation thickness and robustness; *iii*) R2ET proves its generalizability by showing its superiority across diverse data modalities, model structures, and explanation methods such as concept-based and saliency map-based.

## 2 Related Work

**Robust Explanations.** Gradient-based methods are widely used due to their simplicity and efficiency [55], while they lack robustness against small perturbations [23]. Adversarial training (AT) is used to improve the explanation robustness [13, 74]. Alternatively, some works propose replacing ReLU function with softplus [17], training with weight decay [16], and incorporating gradient- and Hessian-related terms as regularizers [17, 88, 91]. Besides, some works propose *post-hoc* explanation methods, such as SmoothGrad [75]. More relevant works are discussed in Appendix C.

**Ranking robustness in IR.** Ranking robustness in information retrieval (IR) is well-studied [106, 26, 104]. Ranking manipulations in IR and explanations are different since 1) In IR, the goal is to distort the ranking of candidates by manipulating *one* candidate or the query. We manipulate input to swap *any* pairs of salient and non-salient features. 2) Ranking in IR is based on the model predictions. However, explanations rely on gradients, and studying their robustness requires second or higher-order derivatives, necessitating an efficient regularizer to bypass intensive computations.

## 3 Preliminaries

**Saliency map explanations.** Let a classification model with parameters $\mathbf{w}$ be $f(\mathbf{x}, \mathbf{w}) : \mathbb{R}^n \to [0, 1]^C$, and $f_c(\mathbf{x}, \mathbf{w})$ is the probability of class $c$ for the input $\mathbf{x}$. Denote a saliency map explanation [1] for $\mathbf{x}$ concerning class $c$ as $\mathcal{I}(\mathbf{x}, c; f) = \nabla_{\mathbf{x}} f_c(\mathbf{x}, \mathbf{w})$. Since $f$ is fixed for explanations, we omit

**w** and fix $c$ to the *predicted* class. We denote $\mathcal{I}(\mathbf{x}, c; f)$ by $\mathcal{I}(\mathbf{x})$, and the $i$-th feature's importance score, e.g., the $i$-th entry of $\mathcal{I}(\mathbf{x})$, by $\mathcal{I}_i(\mathbf{x})$. The theoretical analysis following will mainly based on saliency explanations, and more explanation methods, such as Grad× Inp, SmoothGrad [75], and IG [77], are studied in Sec. 6 and Appendix A.1.2.

**Threat model.** The adversary solves the following problem to find the optimal perturbation $\delta^*$ to distort the explanations without changing the predictions [17]:

$$\delta^* = \arg\max_{\delta: \|\delta\|_2 \leq \epsilon} \text{Dist}(\mathcal{I}(\mathbf{x}), \mathcal{I}(\mathbf{x} + \delta)), \quad \text{s.t.} \quad \arg\max_c f_c(\mathbf{x}) = \arg\max_c f_c(\mathbf{x} + \delta). \quad (1)$$

$\delta$ is the perturbation whose $\ell_2$ norm is not larger than a given budget $\epsilon$. $\text{Dist}(\cdot, \cdot)$ evaluates how different the two explanations are. We tackle constraints by constrained optimization (see B.2.3) and will not explicitly show the constraints.

# 4 Explanation Robustness via Thickness

We first present the rationale and formal definition of *thickness* in Sec. 4.1, then propose *R2ET* algorithm to promote the thickness grounded on theoretical analyses in Sec. 4.2. All omitted proofs and more theoretical discussions are provided in Appendix A.

## 4.1 Ranking explanation thickness

**Quantify the gap.** Given a model $f$ and the associated original explanation $\mathcal{I}(\mathbf{x})$ with respect to an input $\mathbf{x} \in \mathbb{R}^n$, we denote the *gap* between the $i$-th and $j$-th features' importance by $h(\mathbf{x}, i, j) = \mathcal{I}_i(\mathbf{x}) - \mathcal{I}_j(\mathbf{x})$. Clearly, $h(\mathbf{x}, i, j) > 0$ *if and only if* the $i$-th feature has a more positive contribution to the prediction than the $j$-th feature. The magnitude of $h$ reflects the disparity in their contributions. Although the feature importance order varies across different $\mathbf{x}$, for notation brevity, we label the features in a descending order such that $h(\mathbf{x}, i, j) > 0, \forall i < j$, holds for any *original* input $\mathbf{x}$. This assumption will not affect the following analysis.

**Pairwise thickness.** The adversary in Eq. (1) searches for a perturbed input $\mathbf{x}'$ to flip the ranking between features $i$ and $j$ so that $h(\mathbf{x}', i, j) < 0$ for some $i < j$. A simple evaluation metric, such as $\theta = \mathbb{1}[h(\mathbf{x}', i, j) \geq 0]$, can only indicate if the ranking flips at $\mathbf{x}'$, failing to quantify the extent of disarray. Moreover, it is greatly impacted by the choice of $\mathbf{x}'$. Conversely, the *explanation thickness* is defined by the expected gap of the relative ranking of the feature pair $(i, j)$ in a neighborhood of $\mathbf{x}$.

**Definition 4.1** (Pairwise ranking thickness). Given a model $f$, an input $\mathbf{x} \in \mathcal{X}$ and a distribution $\mathcal{D}$ of perturbed input $\mathbf{x}' \sim \mathcal{D}$, the pairwise ranking thickness of the pair of features $(i, j)$ is

$$\Theta(f, \mathbf{x}, \mathcal{D}, i, j) \overset{\text{def}}{=} \mathbb{E}_{\mathbf{x}' \sim \mathcal{D}} \left[ \int_0^1 h(\mathbf{x}(t), i, j) dt \right], \quad \text{where } \mathbf{x}(t) = (1 - t)\mathbf{x} + t\mathbf{x}', t \in [0, 1]. \quad (2)$$

The integration computes the average gap between two features from $\mathbf{x}$ to $\mathbf{x}'$. The expectation over $\mathbf{x}' \sim \mathcal{D}$ makes an overall estimation. For example, $\mathbf{x}'$ can be from a Uniform distribution [88] or the adversarial samples local to $\mathbf{x}$ [97]. The relevant work [97] proposes the boundary thickness to evaluate a model's *prediction* robustness by the distance between two level sets.

**Top-$k$ thickness.** Existing works in general robust ranking [105] propose maintaining the ranking between *every* two entities (features). However, as shown in Fig. 1, only the top-$k$ important features in $\mathcal{I}(\mathbf{x})$ attract human perception the most, and will be delivered to end-users. Thus, we focus on the relative ranking between a top-$k$ salient feature and any other non-salient one.

**Definition 4.2** (Top-$k$ ranking thickness). Given a model $f$, an input $\mathbf{x} \in \mathcal{X}$, and a distribution $\mathcal{D}$ of $\mathbf{x}' \sim \mathcal{D}$, the ranking thickness of the top-$k$ features is

$$\Theta(f, \mathbf{x}, \mathcal{D}, k) \overset{\text{def}}{=} \frac{1}{k(n - k)} \sum_{i=1}^{k} \sum_{j=k+1}^{n} \Theta(f, \mathbf{x}, \mathcal{D}, i, j). \quad (3)$$

**Variants and discussions.** We consider and discuss the following three major variants of thickness.

- One variant of thickness in Eq. (2) is in a *probability* version:

$$\Theta_{\text{Pr}}(f, \mathbf{x}, \mathcal{D}, i, j) \overset{\text{def}}{=} \mathbb{E}_{\mathbf{x}' \sim \mathcal{D}} \left[ \int_0^1 \mathbb{1}[h(\mathbf{x}(t), i, j) \geq 0] dt \right].$$

However, the inherent non-differentiability of the indicator function hinders effective analysis and optimization of thickness. One may replace it with a sigmoid to alleviate the non-differentiability issue. Yet, the use of a sigmoid can potentially lead to vanishing gradient problems.

- By considering the relative positions of the top-$k$ features, one can define an *average precision@k* like thickness $\Theta_{ap} \stackrel{\text{def}}{=} \frac{1}{K} \sum_{k=1}^{K} \Theta(f, \mathbf{x}, \mathcal{D}, k)$, or a *discounted cumulative gain* [33] like thickness $\Theta_{dcg} \stackrel{\text{def}}{=} \sum_{i=1}^{k} \sum_{j=k+1}^{n} \frac{\Theta(f, \mathbf{x}, \mathcal{D}, i, j)}{\log(j-i+1)}$. Other variants for specific properties or purposes can be derived similarly, and we will study these variants deeply in future work.

- Many non-salient features are less likely to be confused with the top-$k$ salient features (see Fig. 8), and treating them equally may complicate optimization. Thus, one may approximate the top-$k$ thickness by $k'$ pairs of features $\sum_{i=1}^{k'} h(\mathbf{x}, k-i, k+i)$, which selects $k'$ distinct pairs with *minimal* $h(\mathbf{x}, i, j)$. We include the case when $k' = k$ denoted with the suffix "-mm" in the experiments.

### 4.2 R2ET: Training for robust ranking explanations

We will *theoretically* (in Sec. 5) and *experimentally* (in Sec. 6) demonstrate that maximizing the ranking explanation thickness in Eq. (3) can make attacks more difficult and thus the explanation more robust. Thus, a straightforward way is to add $\Theta(f, \mathbf{x}, \mathcal{D}, k)$ as a regularizer during training:

$$\min_{\mathbf{w}} \mathcal{L} = \mathbb{E}_{\mathbf{x}} \left[ \mathcal{L}_{cls}(f, \mathbf{x}) - \lambda \Theta(f, \mathbf{x}, \mathcal{D}, k) \right], \tag{4}$$

where $\mathcal{L}_{cls}$ is the classification loss. However, direct optimization of $\Theta$ requires $M_1 \times M_2 \times 2$ backward propagations *per* training sample. $M_1$ is the number of $\mathbf{x}'$ sampled from $\mathcal{D}$; $M_2$ is the number of interpolations $\mathbf{x}(t)$ between $\mathbf{x}$ and $\mathbf{x}'$; and evaluating the gradient of $h(\mathbf{x}, i, j)$ requires at least 2 backward propagations [59]. In response, we consider the bounds of $\Theta$ in Prop. 4.4, and propose **R2ET** which requires *only* 2 backward propagations each time.

**Definition 4.3** (Locally Lipschitz continuity)**.** A function $f$ is $L$-locally Lipschitz continuous if $\|f(\mathbf{x}) - f(\mathbf{x}')\|_2 \le L\|\mathbf{x} - \mathbf{x}'\|_2$ holds for all $\mathbf{x}' \in \mathcal{B}_2(\mathbf{x}, \epsilon) = \{\mathbf{x}' \in \mathbb{R}^n : \|\mathbf{x} - \mathbf{x}'\|_2 \le \epsilon\}$.

**Proposition 4.4.** (Bounds of thickness) *Given an $L$-locally Lipschitz model $f$, for some $L > 0$, pairwise ranking thickness $\Theta(f, \mathbf{x}, \mathcal{D}, i, j)$ is bounded by*

$$h(\mathbf{x}, i, j) - \epsilon * \frac{1}{2} \|H_i(\mathbf{x}) - H_j(\mathbf{x})\|_2 \le \Theta(f, \mathbf{x}, \mathcal{D}, i, j) \le h(\mathbf{x}, i, j) + \epsilon * (L_i + L_j), \tag{5}$$

*where $H_i(\mathbf{x})$ is the derivative of $\mathcal{I}_i(\mathbf{x})$ with respect to $\mathbf{x}$, and $L_i = \max_{\mathbf{x}' \in \mathcal{B}_2(\mathbf{x}, \epsilon)} \|H_i(\mathbf{x}')\|_2$.*

Noticing $\lim_{\|H(\mathbf{x})\|_2 \to 0} \Theta(f, \mathbf{x}, \mathcal{D}, i, j) = h(\mathbf{x}, i, j)$, the objective of the proposed **R2ET**, *Robust Ranking Explanation via Thickness*, is to maximize the gap and minimize Hessian norm:

$$\min_{\mathbf{w}} \mathcal{L} = \mathbb{E}_{\mathbf{x}} \left[ \mathcal{L}_{cls} - \lambda_1 \sum_{i=1}^{k} \sum_{j=k+1}^{n} h(\mathbf{x}, i, j) + \lambda_2 \|H(\mathbf{x})\|_2 \right], \tag{6}$$

where $\lambda_1, \lambda_2 \ge 0$. R2ET with $\lambda_1 = 0$ recovers Hessian norm minimization for smooth curvature [17]. The difference between R2ET in Eq. (6) and vanilla model are two regularizers. $h(\mathbf{x}, i, j) = \mathcal{I}_i(\mathbf{x}) - \mathcal{I}_j(\mathbf{x})$ can be calculated by one time backward propagation over $\mathcal{I}(\mathbf{x})$, and the summations are done by assigning different weights to $\mathcal{I}_i(\mathbf{x})$, e.g., $(n - k)$ for $i <= k$, and $-k$ for $i > k$. $\|H(\mathbf{x})\|_2$ is estimated by the finite difference (calculating the difference of the gradient of $\mathcal{I}(\mathbf{x})$ and $\mathcal{I}(\mathbf{x} + \delta)$), which costs two times backward propagation [52]. Thus, R2ET is at an extra cost of 2 times backward propagation. For comparison, AT (in PGD-style) searches the adversarial samples by $M_2$ (being 40 in [13, 70]) iterations for each training sample in every training epoch.

We connect R2ET to three established robustness paradigms, including certified robustness, AT, and constrained optimization. Intuitively, three methods assess robustness from the adversary's perspective by identifying the worst-case samples and restricting model responses to these cases. Conversely, R2ET is defender-oriented, which preserves models' behavior stably by imposing restrictions on the rate of change, avoiding costly adversarial sample searches.

**Connection to certified robustness.** Prop. 4.5 delineates the minimal budget required for a successful attack. The defense strategy of maximizing the budget can effectively mitigate the arms race by obstructing all attacks. The strategy essentially aligns with R2ET, but is empirically less stable and harder to converge due to the second-order term in the denominator. The proof is based on [28].

**Proposition 4.5.** *For all $\delta$ with $\|\delta\|_2 \leq \min_{\{i,j\}} \frac{h(\mathbf{x},i,j)}{\max_{\mathbf{x}'} \|H_i(\mathbf{x}')-H_j(\mathbf{x}')\|_2}$, it holds $\mathbb{1}[h(\mathbf{x},i,j) > 0]$ $= \mathbb{1}[h(\mathbf{x}+\delta,i,j) > 0]$ for all $(i,j)$ pair, that is all the feature rankings do not change.*

**Connection to adversarial training (AT).** Prop. 4.6 implies that R2ET shares the same goal as AT but employs regularizers to avoid costly computations. The proof in Appendix A.3 is based on [94].

**Proposition 4.6.** *The optimization problem in Eq. (6) is equivalent to the following min-max problem:*

$$\min_{\mathbf{w}} \max_{(\delta_{1,k+1},\ldots,\delta_{k,n})\in\mathcal{M}} \mathbb{E}_{\mathbf{x}} \left[ \mathcal{L}_{cls} - \sum_{i=1}^{k} \sum_{j=k+1}^{n} h(\mathbf{x}+\delta_{i,j},i,j) \right], \tag{7}$$

*where $\delta_{i,j} \in \mathbb{R}^n$ is a perturbation to $\mathbf{x}$ targeting at the $(i,j)$ pair of features. $\mathcal{M}$ is the feasible set of perturbations where each $\delta_{i,j}$ is independent of each other, with $\|\sum_{i,j} \delta_{i,j}\| \leq \epsilon$.*

**Connection to constrained optimization.** Prop. 4.7 shows that R2ET aims to keep the feature in the "correct" positions, e.g., $l_i \mathcal{I}_i \geq G_i$, for the local vicinity of original inputs, thus promoting robustness. First, Eq. (6) can be considered a Lagrange function for the constrained problem in Eq. (8):

$$\min_{\mathbf{w}} \quad \mathbb{E}_{\mathbf{x}} \left[ \mathcal{L}_{cls} + \lambda_2 \|H(\mathbf{x})\|_2 \right], \qquad \text{s.t.} \quad l_i \mathcal{I}_i(\mathbf{x}) \geq G_i, \ \forall i \in \{1,\ldots,n\}, \tag{8}$$

where $G_i \geq 0$ is a predefined margin. $l_i = (n-k)$ for $i \leq k$, and $l_i = -k$ otherwise, which labels features at the top as positive and those at the bottom as negative. The proof is based on [34].

**Proposition 4.7.** *The optimization problem in Eq. (8) is equivalent to the following problem:*

$$\min_{\mathbf{w}} \quad \mathbb{E}_{\mathbf{x}} \left[ \mathcal{L}_{cls} \right] \qquad \text{s.t.} \min_{\delta_i : \|\delta_i\|_2 \leq \epsilon} l_i \mathcal{I}_i(\mathbf{x}+\delta_i) \geq G_i, \ \forall i \in \{1,\ldots,n\}. \tag{9}$$

## 5 Analyses of numerical and statistical robustness

**Numerical stability of explanations.** Different from prior work, we characterize the worst-case complexity of explanation robustness using iterative numerical algorithm for constrained multi-objective optimization. *Threat model.* The attacker aims to swap the ranking of *any* pair of a salient feature $i$ and a non-salient feature $j$, so that the gap $h(\mathbf{x},i,j) < 0$, while does not change the input $\mathbf{x}$ and the predictions $f(\mathbf{x})$ significantly for stealthiness. We assume that the attacker has access to the parameters and architecture of the model $f$ to conduct gradient-based white-box attacks.

**Optimization problem for the attack.** Each $h(\mathbf{x},i,j)$ can be treated as an objective function for the attacker, who however does not know which objective is the easiest to attack. Furthermore, attacking one objective can make another easier or harder to attack due to their dependency on the same model $f$ and the input $\mathbf{x}$. As a result, an attack against a specific target feature pair does not reveal the true robustness of the feature ranking. We quantify the hardness of a successful attack (equivalently, explanation robustness) by an upper bound on the number of iterations for the attacker to flip the *first* (unknown) feature pair. The longer it takes to flip the first pair, the more robust the explanations of $f(\mathbf{x})$ are. Formally, for any given initial input $\mathbf{x}^{(0)}$ ($(p)$ in the superscript means the $p$-th attacking iteration), the attacker needs to iteratively manipulate $\mathbf{x}$ to reduce the gaps $h(\mathbf{x},i,j)$ for all $i \in \{1,\ldots,k\}$ and $j \in \{k+1,\ldots,n\}$ packed in a vector objective function:

$$\text{MOO-Attack}(\mathbf{x},\epsilon): \ \min_{\mathbf{x}}[h_1(\mathbf{x}),\ldots,h_m(\mathbf{x})], \quad \text{s.t.} \ \|f(\mathbf{x}) - f(\mathbf{x}^{(0)})\| \leq \epsilon, \tag{10}$$

where $h_\ell(\mathbf{x}) \stackrel{\text{def}}{=} h(\mathbf{x},i,j)$, with $\ell = 1,\ldots,m$ indexing the $m$ pairs of $(i,j)$.

*Comments:* (*i*) The scalar function $\sum_i \sum_j h(\mathbf{x},i,j)$ is unsuitable for theoretical analyses, since there can be no pair of features flipped at a stationary point of the sum. Simultaneously minimizing multiple objectives identifies when the first pair of features is flipped and gives a pessimistic bound for defenders (and an optimistic bound for attackers). (*ii*) We could adopt the Pareto criticality for convergence analysis [21], but a Pareto critical point indicates that no critical direction (moving in which does not violate the constraint) leads to a joint descent in *all* objectives, rather than that an objective function $h_\ell$ has been sufficiently minimized. (*iii*) Different from the unconstrained case in [19], we constrain the amount of changes in $\mathbf{x}$ and $f(\mathbf{x})$.

We propose a trust-region method, Algorithm 1, to solve MOO-Attack (A.4 for more details). First, we define the merit function $\phi_\ell$ that combines the $\ell$-th objective and the constraint in Eq. (10):

$$\phi_\ell(\mathbf{x}, t) \overset{\text{def}}{=} \|f(\mathbf{x})\| + |h_\ell(\mathbf{x}) - t|. \tag{11}$$

As an approximating model for the trust-region method, the linearization of $\phi(\mathbf{x}, t)$ is

$$l_{\phi_\ell}(\mathbf{x}, t, \mathbf{d}) = \|f(\mathbf{x}) + J(\mathbf{x})^\top \mathbf{d}\| + |h_\ell(\mathbf{x}) + g_\ell(\mathbf{x})^\top \mathbf{d} - t|,$$

where $J(\mathbf{x})$ is the Jacobian of $f$ at $\mathbf{x}$ and $g_\ell(\mathbf{x})$ is the gradient of $h_\ell$. The maximal reduction in $l_{\phi_\ell}(\mathbf{x}, t, 0)$ within a radius of $\Delta > 0$ is

$$\chi_\ell(\mathbf{x}, t) \overset{\text{def}}{=} l_{\phi_\ell}(\mathbf{x}, t, 0) - \min_{\mathbf{d}: \|\mathbf{d}\| \leq \Delta} l_{\phi_\ell}(\mathbf{x}, t, \mathbf{d}). \tag{12}$$

$\mathbf{x}$ is called a critical (approximately critical) point of $\phi_\ell$ when $\chi_\ell(\mathbf{x}, t) = 0$ (or $\leq \epsilon$).

---

**Algorithm 1** Attacking a pair of features

1: **Input**: target model $f$, input $\mathbf{x}$, explanation $\mathcal{I}(\mathbf{x})$, trust-region method parameters $0 < \gamma, \eta < 1$.
2: Set $k = 1$, $\mathbf{x}^{(p)} = \mathbf{x}$, $t_\ell^{(p)} = \|f(\mathbf{x}^{(p)})\| + h_\ell(\mathbf{x}^{(p)}) - \epsilon^{(p)}$.
3: **while** $\min_{1 \leq \ell \leq m} \chi_\ell(\mathbf{x}^{(p)}, t^{(p)}) \geq \epsilon$ **do**
4:    Solve TR-MOO$(\mathbf{x}^{(p)}, \Delta^{(p)})$ to obtain a joint descent direction $\mathbf{d}^{(p)}$ for all $l_{\phi_\ell}$.
5:    Update $\rho_\ell^{(p)}$ using $\mathbf{d}^{(p)}$ for each $\ell = 1, \dots, m$.
6:    **if** $\min_\ell \rho_\ell^{(p)} > \eta$ **then**
7:       Update $t_\ell^{(p+1)}$ and $\mathbf{x}^{(p+1)}$.
8:    **else**
9:       Update $\Delta^{(p+1)} = \gamma \Delta^{(p)}$.
10:    **end if**
11: **end while**

---

The core of Algorithm 1 is TR-MOO$(\mathbf{x}, \Delta)$, whose optimal solution provides a descent direction $\mathbf{d}$ for all linearized merit functions $l_{\phi_\ell}$ (not $h_\ell$ or $\phi_\ell$):

$$\text{TR-MOO}(\mathbf{x}, \Delta) \begin{cases} \min_{\alpha, \mathbf{d}} & \alpha \\ \text{s.t.} & l_{\phi_\ell}(\mathbf{x}, t, \mathbf{d}) \leq \alpha, \forall 1 \leq \ell \leq m, \\ & \|\mathbf{d}\| \leq \Delta. \end{cases}$$

where $\alpha$ is the minimal amount of descent and $\Delta$ is the current radius of search for $\mathbf{d}$.

Different from the local convergence results [20], we provide a global rate of convergence for Algorithm 1 to justify thickness maximization in Theorem 5.1. Specifically, it implies that increasing gaps between salient and non-salient features tend to result in larger $h_{\text{up}} - h_{\text{low}}$, which makes it harder for attackers to alter the rankings, leading to stronger explanation robustness.

**Theorem 5.1.** *Suppose that $f(\mathbf{x})$ and $h_\ell(\mathbf{x})$ are continuously differentiable and $h_\ell$ is bounded. Then Algorithm 1 generates an $\epsilon$-first-order critical point for problem Eq. (10) in at most $\left\lceil (h_{\text{up}} - h_{\text{low}}) \frac{\kappa}{\epsilon^2} \right\rceil$ iterations, where $\kappa$ is a constant independent of $\epsilon$ but depending on $\gamma$ and $\eta$ in Algorithm 1, $h_{\text{up}} \overset{\text{def}}{=} \max_\ell \{\max_{\mathbf{x}} h_1(\mathbf{x}), \dots, \max_{\mathbf{x}} h_m(\mathbf{x})\}$, and $h_{\text{low}} \overset{\text{def}}{=} \min_\ell \{\min_{\mathbf{x}} h_1(\mathbf{x}), \dots, \min_{\mathbf{x}} h_m(\mathbf{x})\}$.*

**Statistical stability of explanations.** We use McDairmid's inequality for dependent random variables and the covering number of the space $\mathcal{F}$ of saliency maps $\mathcal{I}$ in Theorem 5.2. The theorem justifies maximizing the gap by showing a bound on how likely a model can rank some non-salient features higher than some salient features due to perturbation to $\mathbf{x}$. Specifically, for a consistent empirical risk level $R_{0,1,u}^{\text{emp}}$, a larger gap corresponds to a larger $u$, which leads to a smaller covering number $\mathcal{N}(\mathcal{F}, \frac{\epsilon u}{8})$. It indicates a reduced probability of high true risk $R_{0,1}^{\text{true}}$, thereby implying more explanations robustness against perturbations. Importantly, this insight is universal regardless of the approach to achieving the larger gaps. See Appendix A.5 for relevant definitions and details.

**Theorem 5.2.** *Given model $f$, input $\mathbf{x}$, surrogate loss $\phi_u$, and a constant $\epsilon > 0$, for arbitrary saliency map $\mathcal{I} \in \mathcal{F}$ and any distribution $\mathcal{D}$ surrounding $\mathbf{x}$ that preserves the salient features in $\mathbf{x}$,*

$$\Pr \left\{ R_{0,1}^{\text{true}}(\mathcal{I}, \mathbf{x}) \geq R_{0,1,u}^{\text{emp}}(\mathcal{I}, \mathbf{x}) + \epsilon \right\} \leq \exp \left( -\frac{2m\epsilon^2}{\chi} \right) \mathcal{N} \left( \mathcal{F}, \frac{\epsilon u}{8} \right),$$

Table 1: P@$k$ (shown in percentage) of different models (rows) under ERAttack / MSE attack. $k = 8$ for the first three dataset, and $k = 50$ for the rest. **Bold** (and underlines) highlight the winner (and runner-up), † indicates the significant superiority between R2ET winner and non-R2ET winner (pairwise t-test at a 5% significance level). ∗ Est-H has about 4% lower clean AUC than others on BP. Exact-H and SSR only apply to tabular datasets, since computing the exact Hessian and its eigenvalues is extremely expensive.

| Method | Adult | Bank | COMPAS | MNIST | CIFAR-10 | ROCT | ADHD | BP |
|---|---|---|---|---|---|---|---|---|
| # of features | 28 | 18 | 16 | 28*28 | 32*32 | 771*514 | 6555 | 3240 |
| Vanilla | 87.6 / 87.7 | 83.0 / 94.0 | 84.2 / 99.7 | 59.0 / 64.0 | 66.5 / 68.3 | 71.9 / 77.7 | 45.5 / 81.1 | 69.4 / 88.9 |
| WD | 91.7 / 91.8 | 82.4 / 85.9 | 87.7 / 99.4 | 59.1 / 64.8 | 64.2 / 65.6 | 77.2 / 68.9 | 47.6 / 79.4 | 69.4 / 88.6 |
| SP | 97.4 / 97.5 | 95.4 / 95.5 | **99.5**† / **100.0** | 62.9 / 66.9 | 67.2 / 71.9 | 73.9 / 69.5 | 42.5 / 81.3 | 68.7 / 90.1 |
| Est-H | 87.1 / 87.2 | 78.4 / 81.8 | 82.6 / 97.7 | 85.2 / 90.2 | 77.1 / 78.7 | 78.9 / **78.0**† | 58.2 / 83.7 | (75.0 / 91.4)∗ |
| Exact-H | 89.6 / 89.7 | 81.9 / 85.6 | 77.2 / 96.0 | - / - | - / - | - / - | - / - | - / - |
| SSR | 91.2 / 92.6 | 76.3 / 84.5 | 82.1 / 97.2 | - / - | - / - | - / - | - / - | - / - |
| AT | 68.4 / 91.4 | 80.0 / 88.4 | 84.2 / 90.5 | 56.0 / 63.9 | 61.6 / 66.8 | 78.0 / 72.9 | 59.4 / 81.0 | 72.0 / 89.0 |
| R2ET$_{\backslash H}$ | **97.5** / **97.7** | **100.0**† / **100.0**† | 91.0 / 99.2 | 82.8 / 89.7 | 67.3 / 72.2 | **79.4** / 70.9 | 60.7 / 86.8 | 70.9 / 89.5 |
| R2ET-mm$_{\backslash H}$ | 93.5 / 93.6 | 95.8 / 98.2 | 95.3 / 97.2 | 81.6 / 89.7 | 77.7 / **79.4**† | 77.3 / 60.2 | 64.2 / 88.8 | 72.4 / 91.0 |
| R2ET | 92.1 / 92.7 | 80.4 / 90.5 | 92.0 / 99.9 | **85.7** / 90.8 | 75.0 / 77.4 | 79.3 / 70.9 | **71.6**† / **91.3**† | 71.5 / 89.9 |
| R2ET-mm | 87.8 / 87.9 | 75.1 / 85.4 | 82.1 / 98.4 | 85.3 / **91.4**† | **78.0**† / 79.1 | 79.1 / 68.3 | 58.8 / 87.5 | **73.8**† / **91.1**† |

where $\mathcal{N}$ is the covering number of the space $\mathcal{F}$ with radius $\frac{\epsilon u}{8}$ [102]. $\chi$ is the chromatic number of a dependency graph of all pairs of salient and non-salient features [81] for McDairmid's inequality.

# 6 Experiments

This section experimentally validates the proposed defense strategies, R2ET, from various aspects, and Appendix B provides more details and results. For a fair comparison, R2ET operates without any prior knowledge of the "true" feature ranking. At each training iteration, R2ET aims to preserve the feature ranking determined in the immediately preceding iteration.

**Datasets.** We adopt DNNs for three tabular datasets: Bank [53], Adult, and COMPAS [54], and two image datasets, CIFAR-10 [36] and ROCT [22]. ROCT consists of real-world medical images having 771x514 pixels on average, making it comparable in scale to CelebA (178x218), ImageNet (469x387), and MS COCO (640x480). Besides, dual-input Siamese networks [48] are adopted on MNIST [40] and two graph datasets, BP [48] and ADHD [49].

**Evaluation metrics.** Precision@$k$ (P@$k$) [88] is used to quantify the similarity between explanations before and after attacks. We adopt three faithfulness metrics [72, 15] in Appendix B.2.6 to show that explanations from R2ET are *faithful*.

All the models have comparable *prediction* performance. Specifically, the vanilla models achieve AUC of 0.87, 0.64, 0.83, 0.99, 0.87, 0.82, 0.69, and 0.76 on Adult, Bank, COMPAS, MNIST, CIFAR-10, ROCT, BP, and ADHD, respectively. Correspondingly, all other models reported only when their AUC is no less than 0.86, 0.63, 0.83, 0.99, 0.86, 0.82, 0.67, and 0.74, respectively.

**Explanation methods.** Gradient (Grad for short) is used as the explanation method if not specified. We also share findings when adopting *robust* explanation method, e.g., SmoothGrad (SG) [75] and Integrated Gradient (IG) [77], and *concept*-based explanation methods (on ROCT datasets).

## 6.1 Compared methods

We conduct two attacks in the PGD manner [50]: Explanation Ranking attack (**ERAttack**) and **MSE attack**. ERAttack minimizes $\sum_{i=1}^{k} \sum_{j=k+1}^{n} h(\mathbf{x}', i, j)$ to manipulate the ranking of features in explanation $\mathcal{I}(\mathbf{x})$, and MSE attack maximizes the MSE (i.e., $\ell_2$ distance) between $\mathcal{I}(\mathbf{x})$ and $\mathcal{I}(\mathbf{x}')$. The proposed defense strategy **R2ET** is compared with the following baselines.

- **Vanilla**: provides the basic ReLU model trained without weight decay or any regularizer term.
- **Curvature smoothing based methods**: Weight decay (**WD**) [17] implicitly binds Hessian norm by weight decay during training, and Softplus (**SP**) [16, 17] replaces ReLU with Softplus$(x; \rho) = \frac{1}{\rho} \ln(1 + e^{\rho x})$. **Est-H** [17] and **Exact-H** consider the *estimated* (by the finite difference [52]) and

Table 2: P@$k$ of models under ERAttack when adopting Grad / SG / IG as the explanation method.

| Method | Adult | Bank | COMPAS | MNIST | ADHD | BP |
|---|---|---|---|---|---|---|
| Vanilla | 87.6 / 94.0 / **71.8** | 83.0 / 90.0 / 88.1 | 84.2 / 92.9 / 94.7 | 59.0 / 67.9 / 82.8 | 45.5 / 39.0 / 56.9 | 69.4 / 59.6 / 60.8 |
| WD | 91.7 / 97.3 / 55.5 | 82.4 / 91.3 / 85.7 | 87.7 / 97.4 / 99.1 | 59.1 / 68.3 / 83.0 | 47.6 / 42.3 / 57.2 | 69.4 / 61.8 / 63.9 |
| SP | 97.4 / 95.3 / 63.9 | 95.4 / 96.7 / 99.9 | **99.5 / 100.0 / 100.0** | 62.9 / 69.0 / 85.4 | 42.5 / 36.9 / 54.9 | 68.7 / 58.6 / 60.5 |
| Est-H | 87.1 / 93.1 / 61.5 | 78.4 / 87.3 / 82.4 | 82.6 / 89.9 / 89.9 | 85.2 / 87.9 / 89.5 | 58.2 / 48.9 / 54.7 | (75.0 / 63.0 / 58.5) |
| AT | 68.4 / 76.6 / 60.0 | 80.0 / 85.9 / 82.6 | 84.2 / 85.9 / 82.4 | 56.0 / 61.5 / 79.3 | 59.4 / 41.2 / 43.0 | 72.0 / 56.7 / 54.4 |
| R2ET$_{\backslash H}$ | **97.5** / 97.5 / 57.3 | **100.0 / 100.0 / 100.0** | 91.0 / 96.6 / 93.6 | 82.8 / 87.1 / 89.0 | 60.7 / 56.9 / 61.9 | 70.9 / 64.2 / **66.0** |
| R2ET-mm$_{\backslash H}$ | 93.5 / 97.4 / 55.9 | 95.8 / 97.0 / 96.3 | 95.3 / 99.1 / 95.5 | 81.6 / 86.8 / 88.7 | 64.2 / 59.5 / 61.9 | 72.4 / 65.5 / 64.1 |
| R2ET | 92.1 / **99.3** / 54.0 | 80.4 / 88.9 / 84.4 | 92.0 / 99.7 / 100.0 | **85.7 / 88.5 / 90.4** | **71.6 / 67.2 / 65.8** | 71.5 / 64.0 / 65.0 |
| R2ET-mm | 87.8 / 98.6 / 54.2 | 75.1 / 85.1 / 80.3 | 82.1 / 93.5 / 99.5 | 85.3 / 88.3 / 90.1 | 58.8 / 50.1 / 51.4 | **73.8 / 65.6** / 63.9 |

*exact* Hessian norm as the regularizer, respectively. **SSR** [88] sets the largest eigenvalue of the Hessian matrix as the regularizer.

- **Adversarial Training (AT)** [31, 92]: finds $f$ by $\min_f \mathcal{L}_{cls}(f; \mathbf{x} + \delta^*, y)$, where $\delta^* = \arg\max_\delta - \sum_{i=1}^{k} \sum_{j=k+1}^{n} h(\mathbf{x} + \delta, i, j)$.

- **Variants of R2ET**: **R2ET-mm** selects *multiple* $(k)$ distinct $i, j$ with *minimal* $h(\mathbf{x}, i, j)$ as discussed in Sec. 4.1. **R2ET$_{\backslash H}$** and **R2ET-mm$_{\backslash H}$** are the ablation variants of R2ET and R2ET-mm, respectively, without optimizing the Hessian-related term in Eq. (6) ($\lambda_2 = 0$). **Est-H** can be considered an ablation variant of R2ET ($\lambda_1 = 0$).

## 6.2   Overall robustness results

**Attackability of ranking-based explanation.** Table 1 reports the explanation robustness of different models against ERAttack and MSE attacks on all datasets. More than 50% of models achieve at least 90% P@$k$ against MSE attacks, concluding that MSE attack cannot effectively alter the rankings of salient features, even without extra defense (row Vanilla). The ineffective attack method can give a false (over-optimistic) impression of explanation robustness. In contrast, ERAttack can displace more salient features from the top-$k$ positions for most models and datasets, leading to significantly lower P@$k$ values than MSE attack. Intuitively, R2ET performs better against attacks since the attackers (by either MSE attack or ERAttack) are supposed to keep the predictions unchanged. R2ET maintains consistency between model explanations and predictions, which ensures that significant manipulation in the top salient features leads to a detectable change in the model's predictions.

**Effectiveness of R2ET against ERAttacks.** We evaluate the effectiveness of different defense strategies against ERAttack, and similar conclusions can be made with MSE attacks case. (1) The best (highest) top-$k$ of R2ET and its variants across most datasets indicate their superiority in preserving the top salient features. (2) It is counter-intuitive that R2ET$_{\backslash H}$, as an ablation version of R2ET, outperforms R2ET on Adult and Bank. The reason is that R2ET$_{\backslash H}$ has the highest thickness on these datasets (see Table 3 in Sec. 6.3). (3) Overall, the curvature smoothing-based baselines without considering the gaps among feature importance perform unstably and usually badly across datasets. Their inferior performance is exactly consistent with the discussion in Sec 4.2. Specifically, *solely* minimizing Hessian-related terms may marginally contribute to the ranking robustness. (4) We do not compare with many AT baselines [74, 70] but Fast-AT [92], which provides a fairer basis for comparing with R2ET and baselines since they require similar training time and resources. However, AT suffers from unstable robust performance and cannot perform well on most datasets.

**Apply R2ET to other explanation methods.** We demonstrate the efficacy and generalizability of R2ET by adopting the concept-based explanation method, SG and IG, respectively.

Concept-based explanation shows the neurons' maximal activations in a specific model layer. We concentrate on the 512 neurons in the penultimate layer of a ResNet [27] on ROCT. To this end, the most stimulated neurons must be stable under perturbations. We define $\mathcal{I}_i$ as the activation map of the $i$-th penultimate neuron, and derive the objective analogous to Eq. (6). Empirical results in Table 1 (col ROCT) illustrate that R2ET and its variants, again, perform best against ERAttack.

Table 2 reports the results of models trained using Grad but evaluated by SG and IG. The following conclusions can be drawn: (1) Compared to Grad, SG and IG generally promote explanation robustness. Notably, regardless of the explanation method used, R2ET and its variants consistently

achieve the highest P@$k$. (2) Applying Grad to R2ETs usually results in greater robustness than adopting SG/IG to baselines. For example in MNIST, R2ET with Grad attains a P@$k$ of 85%, while applying SG to the baselines (except Est-H) fails to exceed a P@$k$ of 70%. (3) R2ET can generalize and transfer the robustness to explanation methods divergent from those utilized during training.

### 6.3 Understanding thickness and attackability

**Assessing model vulnerability: critical role of thickness.** As discussed in Section 5, the number of attack iterations required to reach the *first* successful flip between salient and non-salient features characterizes the ranking explanation robustness. In Fig. 2, each dot in subplots represents an individual sample $\mathbf{x}$. The left subplot demonstrates a high correlation between the sample's thickness and the attacker's required iterations to manipulate the rankings. This high correlation signifies that samples with greater thickness demand a larger attack budget for a successful manipulation, thereby justifying thickness as a more precise metric of explanation ranking robustness.

To understand why R2ET does not always outperform other models, Table 3 juxtaposes P@$k$ with *dataset*-level thickness, defined as the average thickness of all samples across a dataset. Notably, the model exhibiting optimal explanation robustness (P@$k$) consistently displays the greatest thickness, irrespective of the method employed, aligning with Theorem 5.2.

**Optimal Method Selection.** Thickness has proven its efficacy as an indicator for evaluating explanation ranking robustness, rendering it an apt criterion for method selection. Table 1 indicates a more straightforward way to pick a model: deploy R2ET$_{\backslash H}$ (and R2ET-mm$_{\backslash H}$) for datasets with a limited feature set, and R2ET (and R2ET-mm) for datasets for datasets with more features. This is because distinguishing salient and non-salient features is easier in datasets with fewer features. Conversely, maintaining such distinctions becomes more complex and potentially less efficient as the feature count increases, where reducing the Hessian norm is advantageous, as it complicates the manipulation of feature magnitude.

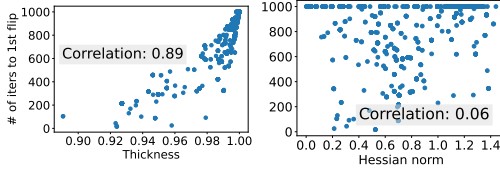

Figure 2: The number of iterations to first flip versus *sample-level* thickness (left) and Hessian norm (right) for R2ET on COMPAS. Each dot represents an individual sample $\mathbf{x}$.

Table 3: P@$k$ under ERAttack / *model*-level thickness. Refer to Table 5 for more results.

| Method | Adult | COMPAS | BP |
|---|---|---|---|
| SP | 97.4 / 0.9983 | **99.5 / 0.9999** | 68.7 / 0.9300 |
| Est-H | 87.1 / 0.9875 | 82.6 / 0.9557 | **75.0 / 0.93563** |
| R2ET$_{\backslash H}$ | **97.5 / 0.9989** | 91.0 / 0.9727 | 70.9 / 0.9271 |
| R2ET | 92.1 / 0.9970 | 92.0 / 0.9865 | 71.5 / 0.9296 |
| R2ET-mm | 87.8 / 0.9943 | 82.1 / 0.9544 | 73.8 / 0.93561 |

### 6.4 Case study: saliency maps visualization

For visual evaluation, Fig. 3 displays models' saliency maps on the ideal testbeds, MNIST and CIFAR-10. On MNIST, Vanilla and WD perform poorly, where about 50% of the top 50 important pixels fell out of top positions under attack. Even worse, the salient pixels identified by Vanilla and WD fail to highlight the digit's spatial patterns (e.g., pixels covering the digits). In contrast, R2ET and R2ET-mm maintain over 90% of salient features encoding the digits' recognizable spatial patterns on the top. Similar trends are observed in CIFAR-10, exemplified by the ship. Vanilla and WD exhibit inferior P@$k$ scores, whereas R2ET and R2ET-mm achieve around 90% P@$k$. Interestingly, all four models identify the front hull of the ship as the significant region for the predictions. However, ERAttack manipulates the explanations of Vanilla and WD to include another region, the wheelhouse, while the critical explanation regions of R2ET and R2ET-mm under attacks remain unaffected. The wheelhouse could be a reason for the ship class, but the inconsistency in explanations due to subtle perturbations raises confusion and mistrust. Fig. 5 provides more results concerning all models.

## 7 Conclusion

We proposed "*explanation ranking thickness*" to measure the robustness of the top-ranked salient features to align with human cognitive capability. We theoretically disclosed the connection between

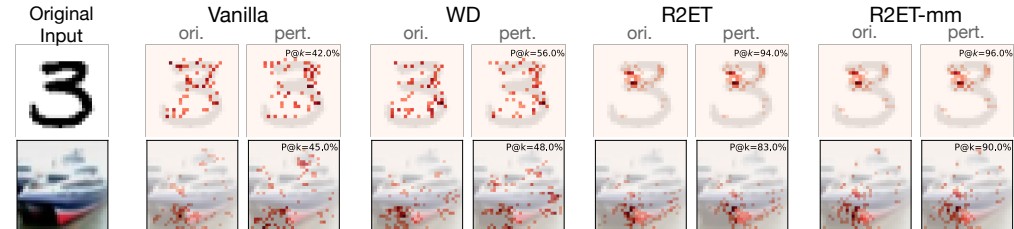

Figure 3: Explanations of original (ori.) and perturbed (pert.) images against ERAttack from MNIST (class digit *3*, $k$=50) and CIFAR-10 (class *ship*, $k$=100). The top $k$ salient pixels are highlighted, and darker colors indicate higher importance. P@$k$ is reported within each subplot.

thickness and a min-max optimization problem, and a global convergence rate of a constrained multi-objective attacking algorithm against the thickness. The theory leads to an efficient training algorithm *R2ET*. On 8 datasets (vectors, images, and graphs), we compared 7 state-of-the-art baselines and 3 variants of R2ET, and consistently confirmed that explanation ranking thickness is a strong indicator of the stability of the salient features. However, R2ET is based on the surrogate loss of thickness, rather than exact thickness, which prevents it from outperforming others all the time. Besides, the theoretical analysis and discussions are based on gradient-based explanation methods. In the future, we plan to further apply R2ET to a broader spectrum of explanation methods. We also plan to investigate scenarios involving highly variable and noisy data and further adjust R2ET to ensure robustness and reliability in more diverse and challenging environments. This paper goals to advance the field of Machine Learning. There are many potential societal consequences of our work, none of which we feel are negative and must be specifically highlighted here.

# 8 Acknowledgments

This material is based upon work supported by the National Science Foundation under Grant Number 2008155. Chao Chen was supported by the National Key Research and Development Program of China (No. 2023YFB3106504), Pengcheng-China Mobile Jointly Funded Project (No. 2024ZY2B0050), and the Natural Science Foundation of China (No. 62476060). Chenghua Guo and Xi Zhang were supported by the Natural Science Foundation of China (No. 62372057). Rufeng Chen and Sihong Xie were supported in part by the National Key R&D Program of China (No. 2023YFF0725001), the Guangzhou-HKUST(GZ) Joint Funding Program (No. 2023A03J0008), and Education Bureau of Guangzhou Municipality. Xiangwen Liao was supported by the Natural Science Foundation of China (No. 62476060).

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

**Appendix A** provides *theoretical* proofs for the propositions and lemmas. **Appendix B** reports *experimental* settings and results. **Appendix C** shows *relevant works* concerning explanation robustness, ranking robustness, and top-$k$ intersection.

## A  Proofs

Proofs concerning Prop. 4.4, 4.5, and 4.6 are in A.1, A.2 and A.3, respectively. As the supplementary of Sec. 5, theoretical analyses on *numerical* and *statistical* robustness are in A.4 and A.5, respectively.

### A.1  Bounds of Ranking Explanation Thickness

#### A.1.1  Bounds of Pairwise Thickness

We denote $\mathbf{x}$ as the target sample, $\|\delta\|_2 \leq \epsilon$ as the perturbation, $\mathbf{x}' = \mathbf{x} + \delta$ as the perturbed input, and $\mathbf{x}(t) = (1-t)\mathbf{x} + t\mathbf{x}', t \in [0, 1]$. Here, $\mathcal{I}(\mathbf{x})$ is not specifically defined as $\nabla_{\mathbf{x}} f(\mathbf{x})$, but an arbitrary explanation method.

**Proposition A.1.** *[Bounds of ranking explanation thickness] Given a $L$-locally Lipschitz model* $f(\mathbf{x})$, *for some* $L \geq \frac{\|\mathbf{x} - \mathbf{x}^*\|_2 \cdot \max_i L_i}{2}$ *where* $\mathbf{x}^* = \arg\max_{\mathbf{x}' \in \mathcal{B}_2(\mathbf{x}, \epsilon)} \frac{\|\mathcal{I}(\mathbf{x}) - \mathcal{I}(\mathbf{x}')\|_2}{\|\mathbf{x} - \mathbf{x}'\|}$, *the ranking explanation thickness for the $(i, j)$ feature pair of a target $\mathbf{x}$ is bounded by*

$$h(\mathbf{x}, i, j) - \epsilon * \frac{1}{2}\|\nabla_{\mathbf{x}}\mathcal{I}_i(\mathbf{x}) - \nabla_{\mathbf{x}}\mathcal{I}_j(\mathbf{x})\|_2 \leq \mathbb{E}_{\mathbf{x}'}\left[\int_0^1 h(\mathbf{x}(t), i, j)dt\right] \leq h(\mathbf{x}, i, j) + \epsilon * (L_i + L_j),$$

*where $H_i(\mathbf{x})$ is the $i$-th column of Hessian matrix of $f$ with respect to the input $\mathbf{x}$, and $L_i = \max_{\mathbf{x}' \in \mathcal{B}_2(\mathbf{x}, \epsilon)} \|\nabla_{\mathbf{x}}\mathcal{I}_i(\mathbf{x}')\|_2$.*

**Lower bound.**

*Proof.* We start from the definition of ranking thickness between $(i, j)$-th features of $\mathbf{x}$ in Eq. (2).

$$
\begin{aligned}
&\int_0^1 h(\mathbf{x}(t), i, j)dt \\
=&\int_0^1 \mathcal{I}_i(\mathbf{x} + t\delta) - \mathcal{I}_j(\mathbf{x} + t\delta)dt \\
\approx&\int_0^1 \mathcal{I}_i(\mathbf{x}) + t\delta^\top \nabla_{\mathbf{x}}\mathcal{I}_i(\mathbf{x}) - \mathcal{I}_j(\mathbf{x}) - t\delta^\top \nabla_{\mathbf{x}}\mathcal{I}_j(\mathbf{x})dt \\
=&\mathcal{I}_i(\mathbf{x}) - \mathcal{I}_j(\mathbf{x}) + \frac{\delta^\top}{2}\nabla_{\mathbf{x}}\mathcal{I}_i(\mathbf{x}) - \frac{\delta^\top}{2}\nabla_{\mathbf{x}}\mathcal{I}_j(\mathbf{x}) \\
\geq&\mathcal{I}_i(\mathbf{x}) - \mathcal{I}_j(\mathbf{x}) - \epsilon * \frac{1}{2}\|\nabla_{\mathbf{x}}\mathcal{I}_i(\mathbf{x}) - \nabla_{\mathbf{x}}\mathcal{I}_j(\mathbf{x})\|_2,
\end{aligned}
\tag{13}
$$

We approximate the gradient at intermediate point $\mathbf{x} + t\delta$ by the Taylor Expansion on line-3. Then the minimum is found with $\delta^* = \arg\min_{\|\delta\|\leq\epsilon} \frac{1}{2}\delta^\top(\nabla_{\mathbf{x}}\mathcal{I}_i(\mathbf{x}) - \nabla_{\mathbf{x}}\mathcal{I}_j(\mathbf{x})) = -\epsilon\frac{\nabla_{\mathbf{x}}\mathcal{I}_i(\mathbf{x}) - \nabla_{\mathbf{x}}\mathcal{I}_j(\mathbf{x})}{\|\nabla_{\mathbf{x}}\mathcal{I}_i(\mathbf{x}) - \nabla_{\mathbf{x}}\mathcal{I}_j(\mathbf{x})\|}$ on line-5. □

**Upper bound.** We introduce two lemmas from [57] and [88] before the proof.

**Lemma A.2.** *If a function $f : \mathbb{R}^n \to \mathbb{R}$ is $L$-locally Lipschitz within $\mathcal{B}_p(\mathbf{x}, \epsilon)$, such that $|f(\mathbf{x}) - f(\mathbf{x}')| \leq L\|\mathbf{x} - \mathbf{x}'\|_p, \forall \mathbf{x}' \in \mathcal{B}_p(\mathbf{x}, \epsilon) = \{\mathbf{x}' : \|\mathbf{x}' - \mathbf{x}\|_p \leq \epsilon\}$, then*

$$L = \max_{\mathbf{x}' \in \mathcal{B}_p(\mathbf{x}, \epsilon)} \|\nabla_{\mathbf{x}'} f(\mathbf{x}')\|_q, \tag{14}$$

*where $\frac{1}{p} + \frac{1}{q} = 1, 1 \leq p, q \leq \infty$.*

**Lemma A.3.** *If a function $f(\mathbf{x})$ is $L$-locally Lipschitz continuous in $\mathcal{B}_2(\mathbf{x}, \epsilon)$, then $\mathcal{I}(\mathbf{x})$ is $K$-locally Lipschitz as well, where $K \leq \frac{2L}{\|\mathbf{x} - \mathbf{x}^*\|}$ and $\mathbf{x}^* = \arg\max_{\mathbf{x}' \in \mathcal{B}_2(\mathbf{x}, \epsilon)} \frac{\|\mathcal{I}(\mathbf{x}) - \mathcal{I}(\mathbf{x}')\|_2}{\|\mathbf{x} - \mathbf{x}'\|_2}$.*

*Proof.* Given an $L$-locally Lipschitz $f(\mathbf{x})$, Lemma A.3 further indicates that $\mathcal{I}_i(\mathbf{x}) : \mathbb{R}^n \to \mathbb{R}$ is locally Lipschitz as well. By adopting $p = q = 2$ on Lemma A.2, the Lipschitz constant is specified by $L_i = \max_{\mathbf{x}' \in \mathcal{B}_2(\mathbf{x}, \epsilon)} \|\nabla_{\mathbf{x}'} \mathcal{I}_i(\mathbf{x}')\|_2$. Formally,

$$|\mathcal{I}_i(\mathbf{x}) - \mathcal{I}_i(\mathbf{x}')| \leq \|\mathcal{I}(\mathbf{x}) - \mathcal{I}(\mathbf{x}')\|_2 \leq L_i \|\mathbf{x} - \mathbf{x}'\|_2.$$

$$
\begin{aligned}
&\int_0^1 h(\mathbf{x}(t), i, j) dt \\
=& h(\mathbf{x}_0', i, j) \\
=& (\mathcal{I}_i(\mathbf{x}_0') - \mathcal{I}_i(\mathbf{x})) - (\mathcal{I}_j(\mathbf{x}_0') - \mathcal{I}_j(\mathbf{x})) + (\mathcal{I}_i(\mathbf{x}) - \mathcal{I}_j(\mathbf{x})) \\
\leq& |\mathcal{I}_i(\mathbf{x}_0') - \mathcal{I}_i(\mathbf{x})| + |\mathcal{I}_j(\mathbf{x}_0') - \mathcal{I}_j(\mathbf{x})| + (\mathcal{I}_i(\mathbf{x}) - \mathcal{I}_j(\mathbf{x})) \\
\leq& L_i \|\mathbf{x}_0' - \mathbf{x}\|_2 + L_j \|\mathbf{x}_0' - \mathbf{x}\|_2 + (\mathcal{I}_i(\mathbf{x}) - \mathcal{I}_j(\mathbf{x})) \\
\leq& \epsilon * (L_i + L_j) + (\mathcal{I}_i(\mathbf{x}) - \mathcal{I}_j(\mathbf{x})).
\end{aligned}
\tag{15}
$$

Based on first mean value theorem, there exists $\mathbf{x}_0'$ within the line segment $\mathbf{x}$ and $\mathbf{x}'$ such that $h(\mathbf{x}_0', i, j) = \int_0^1 h(\mathbf{x}(t), i, j) dt$ on line-2. The Lipschitz constant $L$ of $f$ satisfies $L \geq \frac{\|\mathbf{x} - \mathbf{x}^*\|}{2} * \max_i L_i = \frac{\|\mathbf{x} - \mathbf{x}^*\|}{2} * \max_i \max_{\mathbf{x}' \in \mathcal{B}_2(\mathbf{x}, \epsilon)} \|\nabla_{\mathbf{x}} \mathcal{I}_i(\mathbf{x}')\|_2.$ $\qquad\square$

### A.1.2 Instantiation

Notice that the inequality in Eq. (13) holds for arbitrary explanation methods. Here, we consider different explanation methods (**Grad**, **Grad× Inp**, **SG**, and **IG**) to specify $\mathcal{I}(\mathbf{x})$ and corresponding bounds. We leave *DeepLIFT* [73] and *LRP* [6] to further study, and we do not consider *Grad-CAM* [71] and *Guided Backpropagation* [76] since they are designed only for specific networks.

**Grad.** When adopting Grad as explanation method, where $\mathcal{I}(\mathbf{x}) = \nabla f(\mathbf{x})$ and $\nabla_{\mathbf{x}} \mathcal{I}(\mathbf{x}) = H(\mathbf{x})$, it recovers Prop. 4.4.

$$(\nabla f(\mathbf{x}))_i - (\nabla f(\mathbf{x}))_j - \epsilon * \frac{1}{2} \|H_i(\mathbf{x}) - H_j(\mathbf{x})\|_2 \leq \Theta(f, \mathbf{x}, \mathcal{D}, i, j) \leq h(\mathbf{x}, i, j) + \epsilon * (L_i + L_j), \tag{16}$$

where $H(\mathbf{x})$ is the Hessian matrix, and specifically $L_i = \max_{\mathbf{x}' \in \mathcal{B}_2(\mathbf{x}, \epsilon)} \|H_i(\mathbf{x}')\|_2$.

**SmoothGrad.** When adopting SmoothGrad, where $\mathcal{I}(\mathbf{x}) = \frac{1}{M} \sum_m \nabla f(\mathbf{x} + \beta_m)$ and $\beta_m \sim \mathcal{N}(0, \sigma^2 I)$, we have $\nabla_{\mathbf{x}} \mathcal{I}(\mathbf{x}) = \frac{1}{M} \sum_m H(\mathbf{x} + \beta_m)$. We derive the lower bounds analogy to the derivations on Eq. (13):

$$
\begin{aligned}
&\int_0^1 h(\mathbf{x}(t), i, j) dt \\
=& \frac{1}{M} \sum_m \int_0^1 (\nabla f(\mathbf{x} + t\delta + \beta_m))_i - (\nabla f(\mathbf{x} + t\delta + \beta_m))_j \, dt \\
\geq& \frac{1}{M} \sum_m \left( (\nabla f(\mathbf{x}))_i - (\nabla f(\mathbf{x}))_j - \epsilon * \frac{1}{2} * \|H_i(\mathbf{x}) - H_j(\mathbf{x})\|_2 + \beta_m^T (H_i(\mathbf{x}) - H_j(\mathbf{x})) \right) \\
=& (\nabla f(\mathbf{x}))_i - (\nabla f(\mathbf{x}))_j - \epsilon * \frac{1}{2} * \|H_i(\mathbf{x}) - H_j(\mathbf{x})\|_2 + (\frac{1}{M} \sum_m \beta_m)^T (H_i(\mathbf{x}) - H_j(\mathbf{x})).
\end{aligned}
\tag{17}
$$

Note: when $M$ is large enough, $\frac{1}{M} \sum_m \beta_m \to \mathbf{0}$ since $\beta_m$ is drawn from $\mathcal{N}(0, \sigma^2 I)$.

**Grad $\times$ Inp.** When adopting Grad $\times$ Inp, where $\mathcal{I}(\mathbf{x}) = \nabla f(\mathbf{x}) \odot \mathbf{x}$, we have $\nabla_{\mathbf{x}}\mathcal{I}(\mathbf{x}) = \text{diag}((\nabla f)_1, \ldots, (\nabla f)_n) + H(\mathbf{x})$. The lower bounds derived from Eq. (13) can be further specified:

$$\mathcal{I}_i(\mathbf{x}) - \mathcal{I}_j(\mathbf{x}) - \epsilon * \frac{1}{2}\|\nabla_{\mathbf{x}}\mathcal{I}_i(\mathbf{x}) - \nabla_{\mathbf{x}}\mathcal{I}_j(\mathbf{x})\|_2$$

$$=\mathcal{I}_i(\mathbf{x}) - \mathcal{I}_j(\mathbf{x}) - \epsilon * \frac{1}{2}\|H_i(\mathbf{x}) - H_j(\mathbf{x}) + [0, \ldots, (\nabla f)_i, \ldots, -(\nabla f)_j, \ldots, 0]\|_2$$

$$\geq\mathcal{I}_i(\mathbf{x}) - \mathcal{I}_j(\mathbf{x}) - \epsilon * \frac{1}{2}\left(\|H_i(\mathbf{x}) - H_j(\mathbf{x})\|_2 + \|[0, \ldots, (\nabla f)_i, \ldots, -(\nabla f)_j, \ldots, 0]\|_2\right)$$

$$=\mathcal{I}_i(\mathbf{x}) - \mathcal{I}_j(\mathbf{x}) - \epsilon * \frac{1}{2}\sqrt{(\nabla f)_i^2 + (\nabla f)_j^2} - \epsilon * \frac{1}{2}\|H_i(\mathbf{x}) - H_j(\mathbf{x})\|_2 \qquad (18)$$

$$\geq(\nabla f(\mathbf{x}))_i x_i - (\nabla f(\mathbf{x}))_j x_j - \frac{\epsilon}{2}(\nabla f(\mathbf{x}))_i - \frac{\epsilon}{2}(\nabla f(\mathbf{x}))_j - \epsilon * \frac{1}{2}\|H_i(\mathbf{x}) - H_j(\mathbf{x})\|_2$$

$$=(\nabla f(\mathbf{x}))_i \left(x_i - \frac{\epsilon}{2}\right) - (\nabla f(\mathbf{x}))_j \left(x_j + \frac{\epsilon}{2}\right) - \epsilon * \frac{1}{2}\|H_i(\mathbf{x}) - H_j(\mathbf{x})\|_2$$

$$\approx(\nabla f(\mathbf{x}))_i x_i - (\nabla f(\mathbf{x}))_j x_j - \epsilon * \frac{1}{2}\|H_i(\mathbf{x}) - H_j(\mathbf{x})\|_2.$$

Note: $x_i$ is the $i$-th feature of the input $\mathbf{x}$, and $\epsilon \ll x_i$ since the perturbation is supposed to be neglectable.

**Integrated Grad.** When adopting Integrated Grad, $\mathcal{I}(\mathbf{x}) = (\mathbf{x} - \mathbf{x}^0) \odot \int_{\alpha=0}^{1} \nabla f(\mathbf{x}^0 + \alpha(\mathbf{x} - \mathbf{x}^0))d\alpha$. By setting $\mathbf{x}^0 = \mathbf{0}$, we have $\mathcal{I}(\mathbf{x}) = \mathbf{x} \odot \int_{\alpha=0}^{1} \nabla f(\alpha\mathbf{x})d\alpha$. Similar to the Grad $\times$ Inp case, the lower bound of IG can be derived:

$$\mathcal{I}_i(\mathbf{x}) - \mathcal{I}_j(\mathbf{x}) - \epsilon * \frac{1}{2}\|\nabla_{\mathbf{x}}\mathcal{I}_i(\mathbf{x}) - \nabla_{\mathbf{x}}\mathcal{I}_j(\mathbf{x})\|_2$$

$$\geq \int_{\alpha=0}^{1} (\nabla f(\alpha\mathbf{x}))_i x_i - (\nabla f(\alpha\mathbf{x}))_j x_j - \epsilon * \frac{1}{2}\|H_i(\alpha\mathbf{x}) - H_j(\alpha\mathbf{x})\|_2 d\alpha \qquad (19)$$

$$=\frac{1}{2}\left((\nabla f(\mathbf{x}))_i x_i - (\nabla f(\mathbf{x}))_j x_j - \epsilon * \frac{1}{2}\|H_i(\mathbf{x}) - H_j(\mathbf{x})\|_2\right).$$

### A.1.3 Generalization to Top-$k$ Thickness

Since the inequalities in Eq. (13) and (15) hold for *any* choice of $\mathbf{x}' \in \mathcal{B}_2(\mathbf{x}, \epsilon)$ with a specific $\epsilon$. Thus,

$$h(\mathbf{x}, i, j) - \epsilon * \frac{1}{2}\|\nabla_{\mathbf{x}}\mathcal{I}_i(\mathbf{x}) - \nabla_{\mathbf{x}}\mathcal{I}_j(\mathbf{x})\|_2 \leq \mathbb{E}_{\mathbf{x}'}\left[\int_0^1 h(\mathbf{x}(t), i, j)dt\right] \leq h(\mathbf{x}, i, j) + \epsilon * (L_i + L_j).$$

Furthermore, the bounds of the top-$k$ ranking thickness hold for *any* choice of comparison pairs:

$$\sum_{i,j}\left[h(\mathbf{x}, i, j) - \epsilon * \frac{1}{2}\|\nabla_{\mathbf{x}}\mathcal{I}_i(\mathbf{x}) - \nabla_{\mathbf{x}}\mathcal{I}_j(\mathbf{x})\|_2\right] \leq \mathbb{E}_{\mathbf{x}'}\left[\sum_{i,j}\int_0^1 h(\mathbf{x}(t), i, j)dt\right] \leq \sum_{i,j}\left[h(\mathbf{x}, i, j) + \epsilon * (L_i + L_j)\right].$$

$$(20)$$

Instantiation for the top-$k$ thickness can be derived similarly to those in A.1.2.

### A.2 Connection between R2ET and Certified Robustness

**Proposition A.4.** *For all $\delta$ with $\|\delta\|_2 \leq \min_{\{i,j\}} \frac{h(\mathbf{x},i,j)}{\max_{\mathbf{x}'}\|H_i(\mathbf{x}') - H_j(\mathbf{x}')\|_2}$, it holds $\mathbb{1}[h(\mathbf{x}, i, j) > 0] = \mathbb{1}[h(\mathbf{x} + \delta, i, j) > 0]$ for all $(i, j)$ pair, that is all the feature rankings do not change.*

*Proof.* The proof is adapted from the work [28]. We consider the minimal budget required to achieve $h(\mathbf{x} + \delta, i, j) < 0$, e.g., $\mathcal{I}_i(\mathbf{x} + \delta) - \mathcal{I}_j(\mathbf{x} + \delta) < 0$, given $h(\mathbf{x}, i, j) > 0$. The same proof can be done for $h(\mathbf{x}, i, j) < 0$. Based on the calculus, it holds that

$$\mathcal{I}_i(\mathbf{x} + \delta) = \mathcal{I}_i(\mathbf{x}) + \int_0^1 \langle H_i(\mathbf{x} + t\delta), \delta\rangle dt,$$

$$\mathcal{I}_j(\mathbf{x} + \delta) = \mathcal{I}_j(\mathbf{x}) + \int_0^1 \langle H_j(\mathbf{x} + t\delta), \delta\rangle dt.$$

To achieve $\mathcal{I}_i(\mathbf{x} + \delta) - \mathcal{I}_j(\mathbf{x} + \delta) < 0$,

$$\left(\mathcal{I}_i(\mathbf{x}) + \int_0^1 \langle H_i(\mathbf{x} + t\delta), \delta\rangle dt\right) - \left(\mathcal{I}_j(\mathbf{x}) + \int_0^1 \langle H_j(\mathbf{x} + t\delta), \delta\rangle dt\right) < 0,$$

which implies that

$$\mathcal{I}_i(\mathbf{x}) - \mathcal{I}_j(\mathbf{x}) < \int_0^1 \langle H_j(\mathbf{x} + t\delta), \delta\rangle dt - \int_0^1 \langle H_i(\mathbf{x} + t\delta), \delta\rangle dt.$$

Based on the Hölder's inequality,

$$\mathcal{I}_i(\mathbf{x}) - \mathcal{I}_j(\mathbf{x}) < \|\delta\|_2 \int_0^1 \|H_i(\mathbf{x} + t\delta) - H_j(\mathbf{x} + t\delta)\|_2 dt,$$

Thus, the minimal budget required to flip the ranking between the feature pair $(i, j)$ is

$$\|\delta\|_2 > \frac{\mathcal{I}_i(\mathbf{x}) - \mathcal{I}_j(\mathbf{x})}{\int_0^1 \|H_i(\mathbf{x} + t\delta) - H_j(\mathbf{x} + t\delta)\|_2 dt},$$

$$> \frac{\mathcal{I}_i(\mathbf{x}) - \mathcal{I}_j(\mathbf{x})}{\max_{\mathbf{x}' \in \mathcal{B}_2(\mathbf{x}', \epsilon)} \|H_i(\mathbf{x}') - H_j(\mathbf{x}')\|_2}.$$

Thus, the minimal budget required to flip *any* ranking is $\min_{\{i,j\}} \frac{h(\mathbf{x}, i, j)}{\max_{\mathbf{x}'} \|H_i(\mathbf{x}') - H_j(\mathbf{x}')\|_2}$. $\qquad\square$

## A.3  Connection between AT and R2ET

Following Sec. 3, given a model $f : \mathbb{R}^n \to [0, 1]^C$ and an input $\mathbf{x}$, $\mathcal{I}(\mathbf{x}, c; f) = \nabla_\mathbf{x} f_c(\mathbf{x})$ is the explanation and $\mathcal{I}_i(\mathbf{x})$ is the score for the $i$-th feature. We assume that $\mathcal{I}(\mathbf{x})$ is sorted and top-$k$ ones are salient features. The proof is based on the prior work [94].

The objective of Adversarial Training (AT) for training a model by a min-max game in Sec. 4.2 is:

$$\min_{\mathbf{w}} \max_{(\delta_{1,k+1}, \ldots, \delta_{k,n}) \in \mathcal{M}} \mathcal{L}_{cls} - \sum_{i=1}^{k} \sum_{j=k+1}^{n} h(\mathbf{x} + \delta_{i,j}, i, j). \tag{21}$$

We explicitly show the weight of each $\mathcal{I}_i$, which is $l_i = (n - k)$ if $i \le k$, and $l_i = -k$, otherwise, and rewrite AT's goal:

$$\min_{\mathbf{w}} \max_{(\delta_1, \ldots, \delta_n) \in \mathcal{M}} \mathcal{L}_{cls} - \sum_{i=1}^{n} l_i \mathcal{I}_i(\mathbf{x} + \delta_i), \tag{22}$$

or

$$\max_{\mathbf{w}} \min_{(\delta_1, \ldots, \delta_n) \in \mathcal{M}} -\mathcal{L}_{cls} + \sum_{i=1}^{n} l_i \mathcal{I}_i(\mathbf{x} + \delta_i), \tag{23}$$

where $\delta_i$ is specific to the $i$-th feature of input $\mathbf{x}$.

We will prove the equivalence between R2ET in Eq. (6) and AT in Eq. (21) by proving that

$$\min_{(\delta_1, \ldots, \delta_n) \in \mathcal{M}} \sum_{i=1}^{n} l_i \mathcal{I}_i(\mathbf{x} + \delta_i) \tag{24}$$

is equivalent to

$$\nu = \sum_{i=1}^{n} l_i \mathcal{I}_i(\mathbf{x}) - \epsilon \max_t l_t \|H_t(\mathbf{x})\|_2. \tag{25}$$

*Proof.* Given an $\epsilon$ norm ball $\mathcal{M}_0 \overset{\text{def}}{=} \{\delta \in \mathbb{R}^n : \|\delta\| \le \epsilon\}$, we consider a perturbation set $\mathcal{M}$ where perturbations on *each feature* are independent, but the *aggregation* of perturbations is controlled. Formally, $\mathcal{M}$ satisfies $\mathcal{M}^- \subseteq \mathcal{M} \subseteq \mathcal{M}^+$, where

$$\mathcal{M}^- \overset{\text{def}}{=} \cup_{i=1}^n \mathcal{M}_i^-, \quad \text{where } \mathcal{M}_i^- \overset{\text{def}}{=} \{(\delta_1, \ldots, \delta_n) | \delta_i \in \mathcal{M}_0; \delta_{t \ne i} = \mathbf{0}\};$$

$$\mathcal{M}^+ \overset{\text{def}}{=} \{(\alpha_1 \delta_1, \ldots, \alpha_n \delta_n) | \sum_{i=1}^n \alpha_i = 1; \alpha_i \ge 0, \delta_i \in \mathcal{M}_0, i = 1, \ldots, n\}.$$

$\mathcal{M}^- \subseteq \mathcal{M} \subseteq \mathcal{M}^+$ naturally implies that

$$\min_{(\delta_1,...,\delta_n)\in\mathcal{M}^+} \sum_{i=1}^n l_i \mathcal{I}_i(\mathbf{x}+\delta_i) \leq \min_{(\delta_1,...,\delta_n)\in\mathcal{M}} \sum_{i=1}^n l_i \mathcal{I}_i(\mathbf{x}+\delta_i) \leq \min_{(\delta_1,...,\delta_n)\in\mathcal{M}^-} \sum_{i=1}^n l_i \mathcal{I}_i(\mathbf{x}+\delta_i).$$
(26)

We will prove that $\nu$ is not smaller than the rightmost term and not larger than the leftmost term in Eq. (26).

**To prove** $\nu \geq \min_{(\delta_1,...,\delta_n)\in\mathcal{M}^-} \sum_{i=1}^n l_i \mathcal{I}_i(\mathbf{x}+\delta_i)$**.**

$$\min_{(\delta_1,...,\delta_n)\in\mathcal{M}^-} \sum_{i=1}^n l_i \mathcal{I}_i(\mathbf{x}+\delta_i)$$

$$\leq \min_{(\delta_1,...,\delta_n)\in\mathcal{M}_i^-} \sum_{i=1}^n l_i \mathcal{I}_i(\mathbf{x}+\delta_i)$$

$$= \sum_{i=1}^n l_i \mathcal{I}_i(\mathbf{x}) + \sum_{i=1}^n \min_{(\delta_1,...,\delta_n)\in\mathcal{M}_i^-} l_i H_i(\mathbf{x})^\top \delta_i$$

$$= \sum_{i=1}^n l_i \mathcal{I}_i(\mathbf{x}) + \min_{t\in\{1,...,n\}} \min_{\delta_t\in\mathcal{M}_0} l_t H_t(\mathbf{x})^\top \delta_t$$

$$= \sum_{i=1}^n l_i \mathcal{I}_i(\mathbf{x}) - \epsilon \max_{t\in\{1,...,n\}} l_t \|H_t(\mathbf{x})\|_2.$$

The inequality on line-2 holds since $\mathcal{M}_i^- \subseteq \mathcal{M}^-$. The equality on line-4 holds due to the definition of $\mathcal{M}_i^-$, where only one $\delta_t$ of $(\delta_1,\dots,\delta_n)$ is from $\mathcal{M}_0$ and the rest are all zeros. The equality on line-5 holds by picking $\delta_t = -\epsilon \frac{H_t(\mathbf{x})}{\|H_t(\mathbf{x})\|}$.

**To prove** $\nu \leq \min_{(\delta_1,...,\delta_n)\in\mathcal{M}^+} \sum_{i=1}^n l_i \mathcal{I}_i(\mathbf{x}+\delta_i)$**.**

$$\min_{(\delta_1,...,\delta_n)\in\mathcal{M}^+} \sum_{i=1}^n l_i \mathcal{I}_i(\mathbf{x}+\delta_i)$$

$$= \min_{\sum_i \alpha_i=1,\alpha_i\geq 0,\hat{\delta}_i\in\mathcal{M}_0} \sum_{i=1}^n l_i \alpha_i \mathcal{I}_i(\mathbf{x}+\hat{\delta}_i)$$

$$= \sum_{i=1}^n l_i \mathcal{I}_i(\mathbf{x}) + \min_{\sum_i \alpha_i=1,\alpha_i\geq 0} \sum_{i=1}^n \alpha_i \min_{\hat{\delta}_i\in\mathcal{M}_0} l_i H_i(\mathbf{x})^\top \delta_i$$

$$= \sum_{i=1}^n l_i \mathcal{I}_i(\mathbf{x}) + \min_{t\in\{1,...,n\}} \min_{\hat{\delta}_t\in\mathcal{M}_0} l_t H_t(\mathbf{x})^\top \delta_t$$

$$= \sum_{i=1}^n l_i \mathcal{I}_i(\mathbf{x}) - \epsilon \max_{t\in\{1,...,n\}} l_t \|H_t(\mathbf{x})\|_2.$$

$\square$

## A.4 A multi-objective attacking algorithm and its analysis

We present an algorithm that will terminate in finite iterations, and flip the first pair of salient and non-salient features, or claim a failed attack. The algorithm is based on a trust-region method designed for single non-convex but smooth objective with nonlinear and non-convex equality constraints [9].

Let the output vector $f(\mathbf{x}) = [f_1(\mathbf{x}),\dots,f_C(\mathbf{x})]$. Given $n$ features and top-$k$ salient features, there are $m = k \times (n-k)$ objectives that can be indexed by subscripts $\ell$ so that the objective vector becomes $[h_1(\mathbf{x}),\dots,h_\ell(\mathbf{x}),\dots,h_m(\mathbf{x})]$. Let $\|\|$ be a convex norm with Lipschitz constant 1.

---

**Algorithm 2** Attacking a pair of features

---

**Input**: initial input $\mathbf{x}$, target model $f$, current explanation $\mathcal{I}(\mathbf{x})$, tolerance $\epsilon > 0$, trust-region method parameters $1 > \eta > 0$ and $1 > \gamma > 0$.

Set $k = 1$, $\mathbf{x}^{(p)} = \mathbf{x}$, $t_\ell^{(p)} = \|f(\mathbf{x}^{(p)})\| + h_\ell(\mathbf{x}^{(p)}) - \epsilon^{(p)}$ for each objective $h_\ell$.

**while** $\min_{1 \le \ell \le m} \chi_\ell(\mathbf{x}^{(p)}, t^{(p)}) \ge \epsilon$ **do**

    Solve TR-MOO($\mathbf{x}^{(p)}, \Delta^{(p)}$) to obtain a joint descent direction $\mathbf{d}^{(p)}$ for all linearized merit functions $l_{\phi_\ell}$.

    $\rho_\ell^{(p)} = \frac{\phi_\ell(\mathbf{x}^{(p)}, t^{(p)}) - \phi_\ell(\mathbf{x}^{(p)} + d^{(p)}, t^{(p)})}{l_{\phi_\ell}(\mathbf{x}^{(p)}, t_\ell^{(p)}, 0) - l_{\phi_\ell}(\mathbf{x}^{(p)}, t_\ell^{(p)}, \mathbf{d}^{(p)})}$ for each $\ell = 1, \ldots, m$.

    **if** $\min_\ell \rho_\ell^{(p)} > \eta$ **then**

        $\mathbf{x}^{(p+1)} = \mathbf{x}^{(p)} + \mathbf{d}^{(p)}$.

        $\Delta^{(p+1)} = \Delta^{(p)}$.

        **if** $h_\ell(\mathbf{x}^{(p)}) \ge t_\ell^{(p)}$ **then**

            $t_\ell^{(p+1)} = t_\ell^{(p)} - \phi_\ell(\mathbf{x}^{(p)}, t_\ell^{(p)}) + \phi_\ell(\mathbf{x}^{(p+1)}, t_\ell^{(p)})$.

        **else**

            $t_\ell^{(p+1)} = 2h_\ell(\mathbf{x}^{(p+1)}) - t^{(p)} - \phi_\ell(\mathbf{x}^{(p)}, t_\ell^{(p)}) + \phi_\ell(\mathbf{x}^{(p+1)}, t_\ell^{(p)})$.

        **end if**

    **else**

        $\mathbf{x}^{(p+1)} = \mathbf{x}^{(p)}$.

        $\Delta^{(p+1)} = \gamma \Delta^{(p)}$.

        $t_\ell^{(p+1)} = t_\ell^{(p)}$ for $\ell = 1, \ldots, m$.

    **end if**

**end while**

---

Since we are working with numerical algorithms, an approximately feasible set will be appropriate. Define the following constrained MOO problem

$$\min_{\mathbf{x}} \ [h_1(\mathbf{x}), \ldots, h_\ell(\mathbf{x}), \ldots, h_m(\mathbf{x})], \tag{27}$$

$$\text{s.t.} \ \|f(\mathbf{x}) - f(\mathbf{x}^{(0)})\| \le \epsilon_f,$$

$$\|\mathbf{x} - \mathbf{x}^{(0)}\|_2 \le \epsilon_x$$

In the sequel, the two constraints can be combined such that

$$\|\tilde{f}(\mathbf{x}) - \tilde{f}(\mathbf{x}^{(0)})\| \le \epsilon, \tag{28}$$

for some $\epsilon > 0$, with $\tilde{f}(\mathbf{x}) = [f(\mathbf{x}), \mathbf{x}]$. Therefore, we let $f$ denote $\tilde{f}$ to simplify the notation. Define the domain

$$\mathcal{C}_1 \stackrel{\text{def}}{=} \{\mathbf{x} : \|f(\mathbf{x}) - f(\mathbf{x}^{(0)})\| \le \epsilon\}. \tag{29}$$

**Assumption A.1**: *The constraint function $f(\mathbf{x})$ is continuously differentiable on the domain $\mathbb{R}^n$, and the objective functions $h_\ell$, $\ell = 1, \ldots, m$, are continuously differentiable in the set*

$$\mathcal{C}_2 \stackrel{\text{def}}{=} \mathcal{C}_1 + \mathcal{B}(0, \delta \Delta^{(1)}), \tag{30}$$

*where $\delta > 1$ is a constant, $\Delta^{(1)}$ is the initial radius argument for the trust-region method for multi-objective optimization TR-MOO($\mathbf{x}, \Delta$) to be defined below, and $\mathcal{B}(0, \delta\Delta^{(1)})$ is an open ball centered at 0 of radius $\delta\Delta^{(1)}$.*

**Assumption A.2**: *The objective functions $h_\ell$, $\ell = 1, \ldots, m$ are bounded in the set $\mathcal{C}_1$. Specifically,*

$$h_{\text{low}} \stackrel{\text{def}}{=} \min_\ell \{\min_{\mathbf{x}} h_1(\mathbf{x}), \ldots, \min_{\mathbf{x}} h_m(\mathbf{x})\}, \tag{31}$$

$$h_{\text{up}} \stackrel{\text{def}}{=} \max_\ell \{\max_{\mathbf{x}} h_1(\mathbf{x}), \ldots, \max_{\mathbf{x}} h_m(\mathbf{x})\}. \tag{32}$$

We give an attacking algorithm MOO-attack that can either find the first feature pair to flip in an explanation, or claim that it is impossible to flip any feature pair. The superscript $(p)$, $p = 1, 2, \ldots$ indicates the number of iterations.

The attacking algorithm may not be able to flip any pair of features when $\min_{1 \leq \ell \leq m} \chi_\ell(\mathbf{x}^{(p)}, t^{(p)}) < \epsilon$, but there can be other objective functions that still have $\chi_\ell(\mathbf{x}^{(p)}, t^{(p)}) \geq \epsilon$ and there is a chance to flip the corresponding pairs of features. We will remove any objective function $h_\ell$ with $\chi_\ell(\mathbf{x}^{(p)}, t^{(p)}) < \epsilon$ and return to the while loop to try to flip other pairs of features. If all objective functions are removed at the end, the attacker fails to attack the explanation.

We adapt the theoretical results from [9] to the above multi-objective optimization algorithm to show a global convergence rate for the attacker.

**Lemma A.5.** *Suppose that assumption A.1 holds. If $\mathbf{x}^{(p)} \in \mathcal{C}_1$, then for each linearized merit function $l_{\phi_\ell}$*

$$l_{\phi_\ell}(\mathbf{x}^{(p)}, t_\ell^{(p)}, 0) - l_{\phi_\ell}(\mathbf{x}^{(p)}, t_\ell^{(p)}, \mathbf{d}^{(p)}) \geq \min(\Delta^{(p)}, 1)\chi_\ell(\mathbf{x}^{(p)}, t^{(p)}). \tag{33}$$

Lemma A.5 shows sufficient descent in the linearized merit functions in the direction $\mathbf{d}^{(p)}$. The proof can be found in Lemma 2.1 in [9].

**Lemma A.6.** *Suppose that assumption A.1 holds. In each iteration for $k \geq 1$ in Algorithm 1, the following properties hold:*

$$h_\ell(\mathbf{x}^{(p)}) - t_\ell^{(p)} > 0, \ell = 1, \ldots, m, \tag{34}$$

$$\phi_\ell(\mathbf{x}^{(p)}, t_k) = \epsilon, \ell = 1, \ldots, m, \tag{35}$$

$$|h_\ell(\mathbf{x}^{(p)}) - t_\ell^{(p)}| \leq \epsilon, \ell = 1, \ldots, m, \tag{36}$$

$$\|f(\mathbf{x}^{(p)})\| \leq \epsilon. \tag{37}$$

Lemma A.6 shows that from iteration to iteration during the loop in Algorithm 1, the invariants will be maintained. The proof can be obtained by applying Lemma 2.2 in [9] to each of the objective functions independently.

The last inequality indicates that during the attack, the manipulated input $\mathbf{x}^{(p)}$ remains in the approximate feasible set $\mathcal{C}_1$ and that the prediction by $f$ is not changed. This is important to make the attack stealthy. The second last inequality shows that each objective $h_\ell$ will chase the corresponding target $t_\ell^{(p)}$ over the iterations, so that if $t_\ell^{(p)}$ can be shown to be decreasing sufficiently fast, we can show the convergence of $h_\ell$. Note that different targets $t_\ell^{(p)}$ will move at different speeds, as they are updated independently in the algorithm. Also, a target is not guaranteed to be reduced below zero for $h_\ell$ to become negative and the $\ell$-th pair of features will be flipped.

**Lemma A.7.** *Suppose that assumptions A.1 and A.2 hold. Then with $\min_\ell \chi_\ell(\mathbf{x}^{(p)}, t^{(p)}) \geq \epsilon$ and*

$$\Delta^{(p)} \leq \min_\ell \frac{(1-\eta)\epsilon}{L_{g_\ell} + \frac{1}{2}L_J}, \tag{38}$$

*the condition $\min_\ell \rho_\ell^{(p)} > \eta$ in Algorithm 1 will hold true and $\Delta^{(p+1)} = \Delta^{(p)}$. Furthermore, with $\min_\ell \chi_\ell(\mathbf{x}^{(p)}, t^{(p)}) \geq \epsilon$, we have, for all $p \geq 1$,*

$$\Delta^{(p)} \geq \min\left(\Delta^{(1)}, \min_\ell \frac{(1-\eta)\gamma}{L_{g_\ell} + \frac{1}{2}L_J}\right)\epsilon. \tag{39}$$

Lemma A.7 shows that when the radius $\Delta^{(p)}$ used in TR-MOO$(\mathbf{x}^{(p)}, \Delta^{(p)})$ is small enough, the radius won't be further reduced. Together with Lemma A.5, the linearized merit functions are reduced sufficiently per iteration. The proof is a modification to the proof of Lemma 3.2 in [9], with the derivation done for each of the objective functions $h_\ell$ to guarantee sufficient descent in all merit functions. An interesting observation is that the search radius $\Delta^{(p)}$ is restricted by the objective that has the most rapid change in its gradient $g_\ell$ (characterized by $L_{g_\ell}$). If all objectives are smooth (with small $L_{g_\ell}$), the search radius can be larger and the reduction in the merit functions is larger. The second inequality of the above lemma says that the search radius will have a lower bound across all iterations.

**Lemma A.8.** *There are at most $O(\lceil|\log(\epsilon)|\rceil)$ number of times that $\Delta^{(p)}$ will be reduced by the factor of $\gamma$.*

Lemma A.8 characterizes the times to reduce the search radius. It is because, starting from $\Delta^{(1)}$, once $\Delta^{(p)}$ falls below $\min_\ell \frac{(1-\eta)\epsilon}{L_{g_\ell} + \frac{1}{2}L_J}$ at iteration $p$, there will be no more reduction in future iterations.

**Lemma A.9.** *Suppose that assumptions A.1 and A.2 hold. Whenever $\mathbf{x}^{(p)}$ is updated in Algorithm 1, for each objective function $h_\ell$, both the reductions $\phi_\ell(\mathbf{x}^{(p)}, t_\ell^{(p)}, 0) - \phi_\ell(\mathbf{x}^{(p)}, t_\ell^{(p)}, \mathbf{d}^{(p)})$ and $t_\ell^{(p)} - t_\ell^{(p)}$ are at least*

$$\min\left(\Delta^{(1)}, \min_\ell \frac{(1-\eta)\gamma}{L_{g_\ell} + \frac{1}{2}L_J}\right)\epsilon^2\eta. \tag{40}$$

Lemma A.9 shows a sufficient reduction in the merit functions and the targets. The proof is to use the condition that $\rho_\ell^{(p)} > \eta$ for all $\ell = 1, \ldots, m$, Eq. (33), the condition that $\min_{1 \le \ell \le m} \chi_\ell(\mathbf{x}^{(p)}, t^{(p)}) \ge \epsilon$, and Eq. (39).

Lastly, we present the main global convergence results.

**Theorem A.10.** *Suppose assumptions A.1-A.2 hold. Then Algorithm 2 generates an $\epsilon-$first-order critical point for problem Eq. (27) in at most*

$$\left\lceil (h_{\text{up}} - h_{\text{low}}) \frac{\kappa}{\epsilon^2} \right\rceil \tag{41}$$

*iterations of the while loop in the algorithm, where $\kappa$ is a constant independent of $\epsilon$ but depending on $\gamma$, $\eta$, $L_{h_\ell}$, and $L_J$.*

The proof hinge on the following inequality

$$h_{\text{low}} \le h_\ell(\mathbf{x}^{(p)}) \tag{42}$$

$$\le t_\ell^{(p)} + \epsilon \tag{43}$$

$$\le t_\ell^{(1)} - i_p\kappa_2\epsilon^2 + \epsilon \tag{44}$$

$$\le h_\ell(\mathbf{x}^{(1)}) - i_p\kappa_2\epsilon^2 + \epsilon \tag{45}$$

$$\le h_{\text{up}} - i_p\kappa_2\epsilon^2 + \epsilon \tag{46}$$

where $i_p$ is the number of iterations between 1 and $p$ where $\min_\ell \rho_\ell^{(p')} > \eta$, $1 \le p \le p'$, is true.

$$\kappa_2 = \min\left(\Delta^{(1)}, \min_\ell \frac{(1-\eta)\gamma}{L_{g_\ell} + \frac{1}{2}L_J}\right)\eta. \tag{47}$$

Therefore,

$$i_p \le \left\lceil \frac{h_{\text{up}} - h_{\text{low}} + \epsilon}{\kappa_2\epsilon^2} \right\rceil. \tag{48}$$

Since $O(\lceil |\log(\epsilon)| \rceil)$ grows slower than $1/\epsilon^2$, the overall number of iterations is in the order of $\left\lceil (h_{\text{up}} - h_{\text{low}}) \frac{\kappa}{\epsilon^2} \right\rceil$.

## A.5 Statistical robustness

We will use tools for generalization error analysis on any single input $\mathbf{x}$ to prove that with a high probability, the empirical error of ranking salient features of $\mathbf{x}$ for a fixed model $f$ is not too far away from the true error of ranking the same set of salient features, with respect to random sampling of $\mathbf{x}$ from an arbitrary distribution $\mathcal{D}$ on a support set around $\mathbf{x}$ where the salient features are preserved.

**Roadmap**: We first adopt the McDairmid's inequality for *dependent* variables to show a concentration inequality for ranking errors of a fixed feature ranking function based on the salient score function $\mathcal{I}(\mathbf{x})$. Then we will adopt the standard covering number argument as in [2, 68] to prove uniform convergence that depends on the margin in the empirical ranking loss, therefore justifying the maximization of the gap/thickness between the scores of salient and non-salient features in R2ET.

**Basic definitions**. Given $\mathcal{I}$ and $\mathbf{x}$, define the true and empirical 0-1 risks

$$R_{0,1}^{\text{true}}(\mathcal{I}, \mathbf{x}) = \mathbb{E}_{\mathbf{x}' \sim \mathcal{D}}\left[\frac{1}{m}\sum_{i=1}^{k}\sum_{j=1}^{n}\mathbb{1}[\mathcal{I}_i(\mathbf{x}') < \mathcal{I}_j(\mathbf{x}')]\right], \tag{49}$$

$$R_{0,1}^{\text{emp}}(\mathcal{I}, \mathbf{x}) = \frac{1}{m} \sum_{i=1}^{k} \sum_{j=1}^{n} \mathbb{1}[\mathcal{I}_i(\mathbf{x}) < \mathcal{I}_j(\mathbf{x})]. \tag{50}$$

To relate the risks to the thickness, consider the loss $\phi_u(z)$, $u > 0$, which is similar to the hinge loss

$$\phi_u(z) = \begin{cases} 1 & z < 0, \\ 1 - z/u & 0 \le z < u, \\ 0 & z \ge u. \end{cases} \tag{51}$$

Based on $\phi_u$, we define the surrogate true and empirical risks

$$R_{\phi,u}^{\text{true}}(\mathcal{I}, \mathbf{x}) = \mathbb{E}_{\mathbf{x}' \sim \mathcal{D}} \left[ \frac{1}{m} \sum_{i=1}^{k} \sum_{j=1}^{n} \phi_u(\mathcal{I}_i(\mathbf{x}') - \mathcal{I}_j(\mathbf{x}')) \right], \tag{52}$$

$$R_{\phi,u}^{\text{emp}}(\mathcal{I}, \mathbf{x}) = \frac{1}{m} \sum_{i=1}^{k} \sum_{j=1}^{n} \phi_u(\mathcal{I}_i(\mathbf{x}) - \mathcal{I}_j(\mathbf{x})). \tag{53}$$

Lastly, an upper bound of $\phi_u(z)$ can be defined as a large-margin 0-1 loss: if $z \ge u$, the loss is 0, and otherwise, the loss is 1. The corresponding empirical risk $R_{0,1,u}^{\text{emp}}(\mathcal{I}, \mathbf{x}) = \frac{1}{m} \sum_{i=1}^{k} \sum_{j=1}^{n} \mathbb{1}[\mathcal{I}_i(\mathbf{x}) < \mathcal{I}_j(\mathbf{x}) + u]$ counts how many pairs of salient and non-salient features that are the salient feature is $u$ less salient than the non-salient feature according to $\mathcal{I}$.

**Generalization bound of $R_{\phi,u}^{\text{emp}}(\mathcal{I}, \mathbf{x})$ for a specific $\mathcal{I}$.** As we randomly sample $\mathbf{x}'$ from $\mathcal{D}$, the terms $\mathcal{I}_i(\mathbf{x}')$ and the associated 0-1 and $\phi_u$ losses. The typical McDiarmid's inequality bounds the probability of a function of multiple independent random variables from the expectation of the function. The elements in $R_{\phi,u}^{\text{emp}}$ are not independent. The saliency scores of different features are dependent since they are function of the same model $f$, the input $\mathbf{x}$, and the mechanism that calculate the gradients for $\mathcal{I}$. In [81], the authors generalized the inequality to dependent variables, and we adopt their conclusion as follows:

**Lemma A.11.** *Given prediction model $f$, saliency map $\mathcal{I}$, input $\mathbf{x}$, distribution $\mathcal{D}$ surrounding $\mathbf{x}$, surrogate loss $\phi_u$, and a constant $\epsilon > 0$, we have*

$$\Pr_{\mathbf{x}' \sim \mathcal{D}} \left\{ R_{\phi,u}^{\text{true}}(\mathcal{I}, \mathbf{x}) \ge R_{\phi,u}^{\text{emp}}(\mathcal{I}, \mathbf{x}) + \epsilon \right\} \le 2 \exp\left( -\frac{2m\epsilon^2}{\chi} \right), \tag{54}$$

*where $\chi$ is the chromatic number of the dependency graph of the pairs of random variables $(\mathcal{I}_i, \mathcal{I}_j)$ for any $1 \le i \le k$ and $k + 1 \le j \le n$.*

*Comments:*

- The above dependency graph describes when two pairs of random variables $(\mathcal{I}_i, \mathcal{I}_j)$ (regarded as a random vector) and $(\mathcal{I}_{i'}, \mathcal{I}_{j'})$ (regarded as another random vector) are dependent. In particular, the two nodes $(\mathcal{I}_i, \mathcal{I}_j)$ and $(\mathcal{I}_{i'}, \mathcal{I}_{j'})$ are linked if they are dependent. This graph depends on $\mathcal{I}$, $\mathbf{x}$, and $\mathcal{D}$ and is in general unknown.

- The chromatic number of the dependency graph is upper-bounded by 1 plus the maximum degree of nodes on the dependency graph, which is $1+m$. In the worst case, the bounds in Eq. (54) approximates $2 \exp(-2\epsilon^2)$ as $m$ becomes sufficiently large.

**Uniform convergence of the class of saliency score functions.** Since R2ET searches the optimal $\mathcal{I}$ function from a class $\mathcal{F}$ of such functions given $f$, $\mathcal{I}$, $\mathbf{x}$, and $\mathcal{D}$, $u > 0$, with the preference over $\mathcal{I}$ that has a large gap $u$ so that $\mathcal{I}_i(\mathbf{x}) > \mathcal{I}_j(\mathbf{x}) + u$ for salient feature $i$ and non-salient feature $j$.

**Theorem A.12.** *Given prediction model $f$, input $\mathbf{x}$, surrogate loss $\phi_u$, and a constant $\epsilon > 0$, for arbitrary saliency map $\mathcal{I} \in \mathcal{F}$ and any distribution $\mathcal{D}$ surrounding $\mathbf{x}$ that preserves the salient features in $\mathbf{x}$, we have*

$$\Pr_{\mathbf{x}' \sim \mathcal{D}} \left\{ R_{0,1}^{\text{true}}(\mathcal{I}, \mathbf{x}) \ge R_{0,1,u}^{\text{emp}}(\mathcal{I}, \mathbf{x}) + \epsilon \right\} \le 2\mathcal{N}(\mathcal{F}, \frac{\epsilon u}{8}) \exp\left( -\frac{2m\epsilon^2}{\chi} \right), \tag{55}$$

*where $\mathcal{N}(\mathcal{F}, \frac{\epsilon u}{8})$ is the covering number of the functional space $\mathcal{F}$ with radius $\frac{\epsilon u}{8}$ [102].*

*Comments:*

- The proof is standard and a similar proof can be found in [68].

- With a larger $u > 0$, the empirical risk $R_{0,1,u}^{\text{emp}}(\mathcal{I}, \mathbf{x})$ will increase as more pairs of salient and non-salient features will not have their saliency score larger than this gap. On the other hand, the covering number $\mathcal{N}(\mathcal{F}, \frac{\epsilon u}{8})$ will decrease as $u$ increases as a larger radius can cover more $\mathcal{I} \in \mathcal{F}$. This represents a model selection problem.

- Last but not least important, with a larger gap $u$ that R2ET can optimize $\mathcal{I}$, with a random perturbation of $\mathbf{x}$ to $\mathbf{x}'$ that has the same saliency features, the true risk (representing how bad the true explanations can be perturbed) will be low.

# B   Details of Experiments

We provide detailed experimental settings in B.1, and comprehensive results in B.2. Specifically, correlation exploration (B.2.1), case study (B.2.2), constrained optimizations (B.2.3), ablation studies (B.2.5), accuracy and explanation robustness trade-off (B.2.4), more explanation methods (**??**), and faithfulness evaluations (B.2.6).

## B.1   Experimental Settings

### B.1.1   Datasets and Target Model Structure

**Tabular data.** Three tabular datasets are included: Adult, Bank [53] and COMPAS [54]. We divide each tabular dataset into training, validation, and test portions at a ratio of $70 : 15 : 15$, respectively. We follow [11] to binarize and map the original inputs mixing with strings and numbers to 28-dim, 18-dim, and 16-dim feature spaces, respectively.

**Image data.** We adopt ResNet18 [27] for CIFAR-10 [36]. As for MNIST [40], we adopt an SN, where the embedding model is a classic CNN, consisting of two convolutional layers with 3*3 kernels followed by max-pooling and three fully-connected layers. Then We use cosine similarity as the similarity metric, and classifier is a single-layer linear model taking the outputs from the embedding models as inputs. To construct pairing samples, we randomly sample 2400 pairs of images with digits *3* and *8* from training images as the training set, and 300 and 600 pairs from test images as the validation and test sets, respectively.

**Graph data.** We apply SNs to two graph datasets, BP [48] and ADHD [49], where each brain network comprises 82 and 116 nodes, respectively. We pair any two training graphs as the training set. To simulate real medical diagnosis (by comparing a new sample with those in the database), each validation (testing) pair consists of a training graph and a validation (test) graph. The embedding model consists of a two-layer GCN, where the first layer maps inputs to a 256-dimension hidden space following ReLU, and then maps to a 128-dimension embedding space. Then it adopts a mean pooling to aggregate node features to graph-level one. The cosine similarity is used to measure the similarity between two embeddings. The classifier is a single-layer linear model taking the outputs from the corresponding embedding models as inputs.

### B.1.2   Methodology of Training Target Models

We train all the models from scratch, except that ResNet is retrained from the Vanilla model. The model with a relatively high cAUC and the highest P@$k$ on the validation set will be adopted as the outstanding model for further evaluation. Specifically, We choose the last model with cAUC higher than a *threshold* on the validation set. Then the outstanding model is the one with the highest P@$k$ out of these high cAUC models on the validation set.

**Details of training by R2ET.** While incorporating a priori information could enhance R2ET's performance, we do not furnish the model with such information to ensure a fair comparison among training methods. 1) **Training from scratch**: The model incrementally develops an understanding of feature importance. At the $t$-th training iteration, it aims to preserve the feature ranking from the $(t-1)$ iteration. This iterative refinement stabilizes the model's perception of feature importance without any predefined ranking knowledge. 2) **Training from a Vanilla Model**: We leverage the established model's feature ranking as a baseline. Given that the vanilla model has achieved

| Description | Values | | | |
|---|---|---|---|---|
| | Tabular | MNIST | CIFAR-10 | Graphs |
| base model architecture* | 2-layer MLP | SN with CNN | ResNet | SN with 2-layer GCN |
| hidden dimension | 32 | 3*3 kernel | [27] | 256 |
| $k$ number of features | 8 | 100 | 100 | 50 |
| $k'$ for R2ET-mm | 8 | 100 | 100 | 20 |
| maximal training epochs | 300 | 100 | 10 (retrain) | 10 |
| learning rate | $10^{-2}$ | $10^{-3}$ | $10^{-2}$ | $10^{-4}$ |
| early stopping | 30 | N/A | N/A | N/A |
| maximal attack iteration | 1000 | 100 | 100 | 100 |
| perturbation per iteration | $10^{-3}$ | $10^{-2}$ | $5*10^{-2}$ | $10^{-2}$ |
| $\kappa$ for Est-H | $\{10^{-6}, 10^{-5}, \ldots, 10^{-2}\}$ | | | |
| $\rho$ for SP | $\{0.5, 1, 5, 10, 100\}$ | | | |
| weight-decay for WD | $\{5*10^{-5}, 5*10^{-4}, 5*10^{-3}, 10^{-2}, 5*10^{-2}\}$ | | | |
| $\lambda$ for single regularization methods | $\{0.01, 0.1, 1, 5, 10, 100\}$ | | | |

Table 4: (Hyper)-parameters used in experiments. $*$ "SN" means Siamese Networks for dual input.

satisfactory performance (AUC), its explanation ranking is a reliable reference. The goal of R2ET is to maintain this inherited feature ranking during retraining.

**Selections of hyperparamters and parameters.** Table 4 shows the default (hyper)-parameters adopted in the experiments. Attacks are conducted in a PGD-style [50], and the infinity norm of the perturbations in each iteration is restricted to no more than 5 for images and 0.2 for graphs. For R2ET, $\lambda_1$ and $\lambda_2$ are selected as the same as the best $\lambda$ for R2ET$_{\setminus H}$ and est-H, respectively. Alternatively, we simply set $\lambda_1 = \lambda_2 = \lambda$, and $\lambda$ can be drawn from $\{0.01, 0.1, 1, 5, 10, 100\}$.

When adopting SmoothGrad (SG) [75] as explanation method, where $\mathcal{I}_{SG}(\mathbf{x}) = \frac{1}{M} \sum_m^M \mathcal{I}(\mathbf{x} + \beta_m)$ and $\beta_m \sim \mathcal{N}(0, \sigma^2 I)$, $M$ is set 50 for all datasets, and $\sigma^2 = 0.5, 25.5^2, 0.01$ for tabular, images, and graphs, respectively. When adopting Integrated Gradients (IG) [77], where $\mathcal{I}_{IG}(\mathbf{x}) = (\mathbf{x} - \mathbf{x}^0) \odot \int_{\alpha=0}^1 \nabla_{\mathbf{x}} f(\mathbf{x}^0 + \alpha(\mathbf{x} - \mathbf{x}^0)) d\alpha$. In practice, we set $\mathbf{x}^0 = \mathbf{0}$, and approximate the integration by interpolating 100 samples between $\mathbf{x}^0$ and $\mathbf{x}$.

**Running environment.** We mainly conduct experiments for three tabular data and CIFAR-10 on the following two machines. Both come with a 16-core Intel Xeon processor and four TITAN X GPUs. One installs 16.04.3 Ubuntu with 3.8.8 Python and 1.7.1 PyTorch, and the other installs 18.04.6 Ubuntu with 3.7.6 Python and 1.8.1 PyTorch. The MNIST and graph datasets are run on a machine with two 10-core Intel Xeon processors and five GeForce RTX 2080 Ti GPUs, which installs 18.04.3 Ubuntu with 3.9.5 Python and 1.9.1 PyTorch.

### B.1.3 Saliency Maps of Target Models

We use the absolute gradient values as the explanation for tabular datasets. For MNIST, we use the normalized absolute gradient values as the explanations. For CIFAR-10, the sum of the absolute gradient values is used as the explanation. For graph data, we use element-wise multiplication of the gradient and the input as the explanation [1].

### B.1.4 Evaluation Metrics

We adopt precision@$k$ to evaluate the explanation *robustness* by quantifying the similarity between $\mathcal{I}(\mathbf{x})$ and $\mathcal{I}(\mathbf{x}')$; AUCs and sensitivity to evaluate the model's *classification performance* (see B.2.3 and B.2.4); and DFFOT, COMP, and SUFF to evaluate the explanation *faithfulness* (see B.2.6).

• **Precision@$k$ (P@$k$).** It is widely used to evaluate the robustness of explanations [23]. Formally,

$$\text{P@}k(\mathcal{I}(\mathbf{x}), \mathcal{I}(\mathbf{x}')) = \frac{|\mathcal{I}(\mathbf{x})_{[k]} \cap \mathcal{I}(\mathbf{x}')_{[k]}|}{k},$$

where $\mathcal{I}(\mathbf{x})_{[k]}$ is the set of the $k$ most important features of the explanation $\mathcal{I}(\mathbf{x})$. $|\cdot|$ counts the number of elements.

- **Clean AUC (cAUC) and adversarial AUC (aAUC).** Besides robust explanations, the model needs robust predictions. Thus, we adopt cAUC and aAUC to measure the model's classification performance before and after the attack, respectively.
- **Sensitivity (Sen).** Since attacks can be detected by checking the consistency of predictions [93], Sen measures the ratio at which classification results change after an attack for models.
- **Decision Flip - Fraction of Tokens (DFFOT)** [72] measures the minimum fraction of important features to be removed to flip the prediction. A *lower* DFFOT indicates a more faithful explanation.

$$\text{DFFOT} = \min_k \frac{k}{n}, \quad \text{s.t.} \ \arg\max_c f_c(\mathbf{x}) \neq \arg\max_c f_c(\mathbf{x}_{[\backslash k]}),$$

where $n$ is the number of features, and $\mathbf{x}_{[\backslash k]}$ is the perturbed input whose top-$k$ important features are removed.
- **Comprehensiveness (COMP)** [15] measures the changes of predictions before and after removing the most important features. A *higher* COMP means a more faithful explanation.

$$\text{COMP} = \frac{1}{\|K\|} \sum_{k \in K} |f_c(\mathbf{x}) - f_c(\mathbf{x}_{[\backslash k]})|,$$

where $K$ is $\{1, \ldots, n\}$ for tabular data, and $\{1\% * n, 5\% * n, 10\% * n, 20\% * n, 50\% * n\}$ for images and graphs. The same setting of $K$ is adopted for SUFF.
- **Sufficiency (SUFF)** [15] measures the change of predictions if only the important tokens are preserved. A *lower* SUFF means a more faithful explanation.

$$\text{SUFF} = \frac{1}{\|K\|} \sum_{k \in K} |f_c(\mathbf{x}) - f_c(\mathbf{x}_{[k]})|,$$

where $\mathbf{x}_{[k]}$ is the perturbed input with only top-$k$ important features.

### B.1.5 Introduction to Siamese Network

An SN predicts whether two samples, $\mathbf{x}^s$ and $\mathbf{x}^t$, are from the same class. The SN uses a network to embed each input, respectively, and measures their similarity by a similarity function. The prediction for class 1 (two samples from the same class), $\Pr(y = 1 | \mathbf{x}^s, \mathbf{x}^t)$, is

$$f_1^{SN}(\mathbf{x}^s, \mathbf{x}^t; \mathbf{w}) = \text{sim}(\text{emb}(\mathbf{x}^s; \mathbf{w}_0), \text{emb}(\mathbf{x}^t; \mathbf{w}_0); \mathbf{w}_1), \tag{56}$$

and $f_0^{SN} = 1 - f_1^{SN}$ for class 0. $\text{sim}(\cdot, \cdot; \mathbf{w}_1)$ is a similarity function, such as the cosine similarity, and $\text{emb}(\cdot; \mathbf{w}_0)$ is an embedding network, such as GNN. $\mathbf{w} = \{\mathbf{w}_0, \mathbf{w}_1\}$ is the SN's parameter set. The ground truth used in SN is $y^{st} = \mathbb{1}[y^s = y^t]$, where $y^s$ is the label of $\mathbf{x}^s$.

To predict a single input, one can further train a classifier $f^{CL}$ based on the embedding part of the SN. For example, $f^{CL}(\mathbf{x}) = \text{cls}(\text{emb}(\mathbf{x}; \mathbf{w}_0'); \mathbf{w}_2)$, where $\mathbf{w}_0'$ can be the same as or retrained from $\mathbf{w}_0$ in Eq. (56), and $\text{cls}(\cdot; \mathbf{w}_2)$ is a classifier such as a linear model with the parameter set $\mathbf{w}_2$.

## B.2 Additional Experimental Results

### B.2.1 Settings for Correlation Experiments and More Results

We explore the correlation between the manipulation epochs, thickness, and Hessian norm concerning different models on various datasets. Specifically, as shown in Fig. 4, we consider different datasets and methods, such as Vanilla, est-H, and R2ET. Since images and graphs have more features than the three tabular datasets, almost all adversarial samples will swap feature pairs under the first epoch of attack. Alternatively, we set the manipulation epoch metric for graph and image datasets to record the first epoch where P@$k$ drops below 0.8. We do not plot the results on Bank, where R2ET achieves 100% P@$k$. Fig. 4 presents the correlations between manipulation epochs and the other four metrics. Besides thickness measured by adversarial samples and Hessian norm as used in Sec. 6.3, we additionally show the thickness measured by either the average or minimal of a few Gaussian samples to mimic *random noise* [106]. Although the thickness evaluated by Gaussian distribution is not specific to adversarial attacks, the corresponding correlations are much higher than between the manipulation epoch and Hessian norm.

The dataset-level thickness is defined as the average thickness over all the samples. Table 5 reports P@$k$ and dataset-level thickness. Clearly, the models with high P@$k$ performance usually have a relatively large thickness as well, and vice versa.

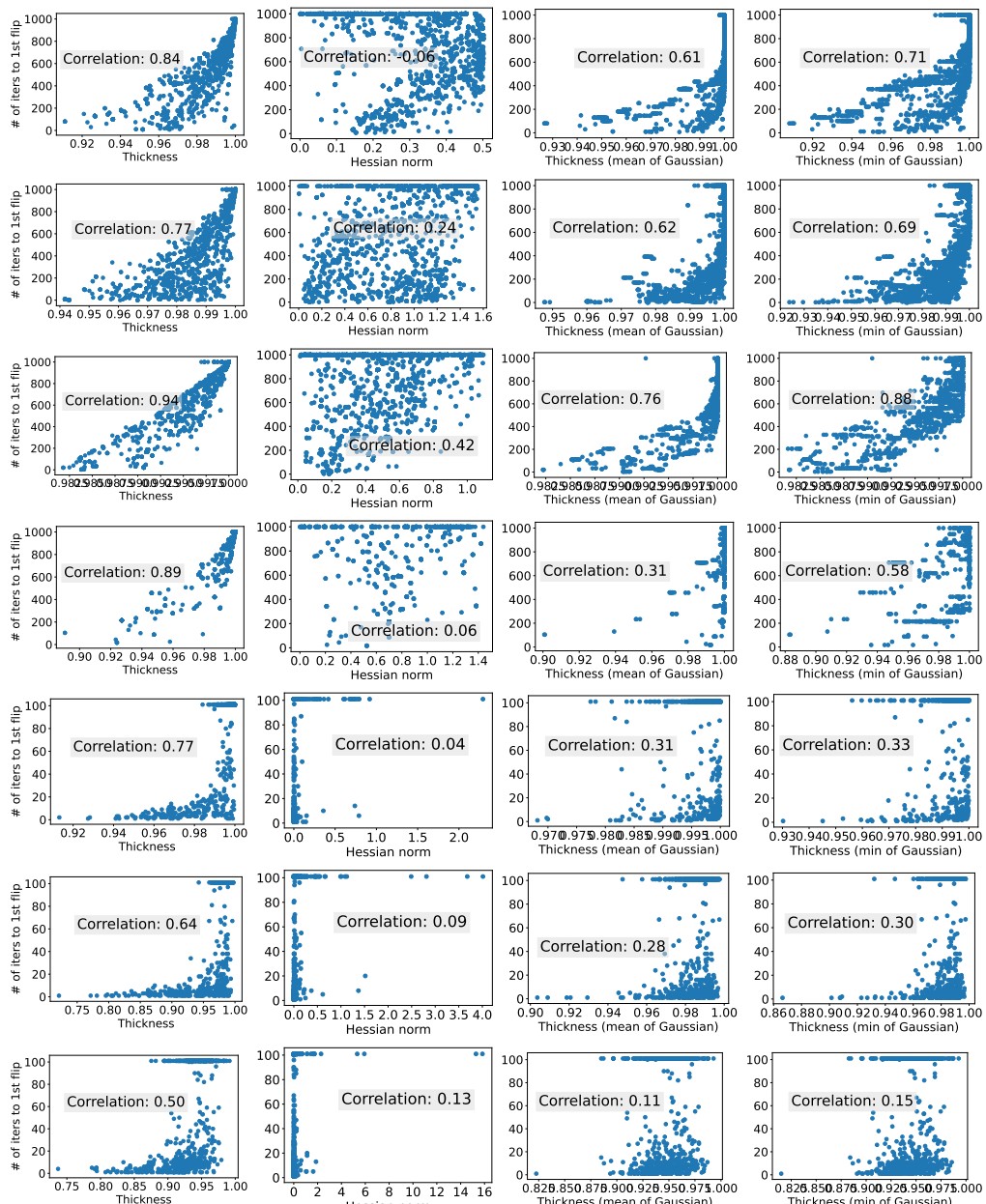

Figure 4: We show the correlation between the manipulation epoch and other metrics, including thickness evaluated by adversarial samples, Hessian norm, and thickness evaluated by (mean and min of) Gaussian samples for different models in various datasets. From top to bottom: Vanilla, est-H models on Adult, respectively. R2ET models on Adult, COMPAS, MNIST, ADHD, and BP, respectively.

Table 5: **P@k** (shown in percentage) of different robust models (rows) under ERAttack and the values of **dataset-level thickness**.

| Method | Adult | Bank | COMPAS | MNIST | ADHD | BP |
|---|---|---|---|---|---|---|
| Vanilla | 87.6 / 0.9889 | 83.0 / 0.9692 | 84.2 / 0.9533 | 59.0 / 0.9725 | 45.5 / 0.9261 | 69.4 / 0.9282 |
| WD | 91.7 / 0.9960 | 82.4 / 0.9568 | 87.7 / 0.9769 | 59.1 / 0.9732 | 47.6 / 0.9343 | 69.4 / 0.9298 |
| SP | 97.4 / 0.9983 | 95.4 / 0.9978 | **99.5** / **0.9999** | 62.9 / 0.9771 | 42.5 / 0.9316 | 68.7 / 0.9300 |
| Est-H | 87.1 / 0.9875 | 78.4 / 0.9583 | 82.6 / 0.9557 | 85.2 / 0.9948 | 58.2 / 0.9578 | **75.0** / **0.9356** |
| Exact-H | 89.6 / 0.9932 | 81.9 / 0.9521 | 77.2 / 0.9382 | - / - | - / - | - / - |
| SSR | 91.2 / 0.9934 | 76.3 / 0.9370 | 82.1 / 0.9549 | - / - | - / - | - / - |
| AT | 68.4 / 0.9372 | 80.0 / 0.9473 | 84.2 / 0.9168 | 56.0 / 0.9639 | 59.4 / 0.9597 | 72.0 / 0.9342 |
| R2ET$_{\backslash H}$ | **97.5** / **0.9989** | **100.0** / **1.0000** | 91.0 / 0.9727 | 82.8 / **0.9949** | 60.7 / 0.9588 | 70.9 / 0.9271 |
| R2ET-mm$_{\backslash H}$ | 93.5 / 0.9963 | 95.8 / 0.9874 | 95.3 / 0.9906 | 81.6 / 0.9942 | 64.2 / 0.9622 | 72.4 / 0.9342 |
| R2ET | 92.1 / 0.9970 | 80.4 / 0.9344 | 92.0 / 0.9865 | **85.7** / **0.9949** | **71.6** / **0.9731** | 71.5 / 0.9296 |
| R2ET-mm | 87.8 / 0.9943 | 75.1 / 0.9102 | 82.1 / 0.9544 | 85.3 / 0.9948 | 58.8 / 0.9588 | 73.8 / **0.9356** |

### B.2.2 More Results for Case Study

Fig. 5 shows the saliency maps for case studies concerning all the baselines and R2ETs.

### B.2.3 Constrained Optimization for Attacks

We show the general form of adversarial attacks by a constrained optimization framework for explanations in Sec. 3. Since defenders can catch adversarial attacks if there are *any* changes in predictions, the attackers must keep *all* $S$ predictions unchanged during the attacks, where $S = 3$ for SNs and $S = 1$ for DNNs. We denote the primary objective in Eq. (1) manipulating the explanations as $g_0$, and constraints for small prediction changes (measured by KL-divergence) as $g_s, 1 \le s \le S$. Naturally, we construct a Lagrangian function with the non-negative multiplier $\boldsymbol{\gamma} = [\gamma_0, \gamma_1, \ldots, \gamma_S] \in \mathbb{R}_+^{S+1}$:

$$\mathcal{L}(\mathbf{x}, \boldsymbol{\gamma}) = \gamma_0 g_0(\mathbf{x}) + \sum_{s=1}^{S} \gamma_s g_s(\mathbf{x}). \tag{57}$$

Some previous works manually set $\boldsymbol{\gamma}$ as a hyperparameter by experience [16], dismissing the relatives among $g_s$. Instead, we care about the unsatisfied constraints with higher weights by uplifting $\gamma_s$ with larger $g_s$. Thus, we adopt the following methods to update both primal variables $\mathbf{x}$ and dual variables $\boldsymbol{\gamma}$ [66, 11]:

- **Gradient descent ascent (GDA)** solves the nonconvex-concave minimax problems [43] by

$$\mathbf{x} \leftarrow \mathbf{x} - \eta_x \frac{\partial \mathcal{L}}{\partial \mathbf{x}}, \quad \boldsymbol{\gamma} \leftarrow \boldsymbol{\gamma} + \eta_\gamma \frac{\partial \mathcal{L}}{\partial \boldsymbol{\gamma}}, \tag{58}$$

  where $\eta_x$ and $\eta_\gamma$ are the learning rates. Notice that $\frac{\partial \mathcal{L}(\mathbf{x}, \gamma)}{\partial \gamma_s} = g_s, \forall s \in \{1, \ldots, S\}$, and $\gamma_0$ is passively updated by the normalization $\sum_{s=0}^{S} \gamma_s = 1$ at the end of each iteration.

- **Hedge** is an incarnation of Multiplicative Weights algorithm that updates $\boldsymbol{\gamma}$ using exponential factors [5]. In each iteration, we normalize $\boldsymbol{\gamma}$ such that $\sum_{s=0}^{S} \gamma_s = 1$, and then update $\boldsymbol{\gamma}$ by

$$\boldsymbol{\gamma} \leftarrow \boldsymbol{\gamma} \odot \exp^{\eta_{Hedge}[g_0, g_1, \ldots, g_S]}, \tag{59}$$

  where $\eta_{Hedge}$ is the learning rate, $\odot$ is the element-wise multiplication, and $\exp$ is exponential operation.

- Another way to update weights $\boldsymbol{\gamma}$ is to solve a **quadratic programming (QP)** problem

$$\max_{\boldsymbol{\gamma}} -\frac{1}{2} \| \sum_{s=0}^{S} \gamma_s \nabla g_s(\mathbf{x}) \|^2, \quad \text{s.t.} \sum_{s=0}^{S} \gamma_s = 1, \quad \gamma_s \ge 0, \forall s \in \{0, \ldots, S\}. \tag{60}$$

We also include **unconstrained** attack without any constraints, and **no update** that fixes the weights to $\boldsymbol{\gamma} = \frac{1}{S+1} \mathbf{1}$.

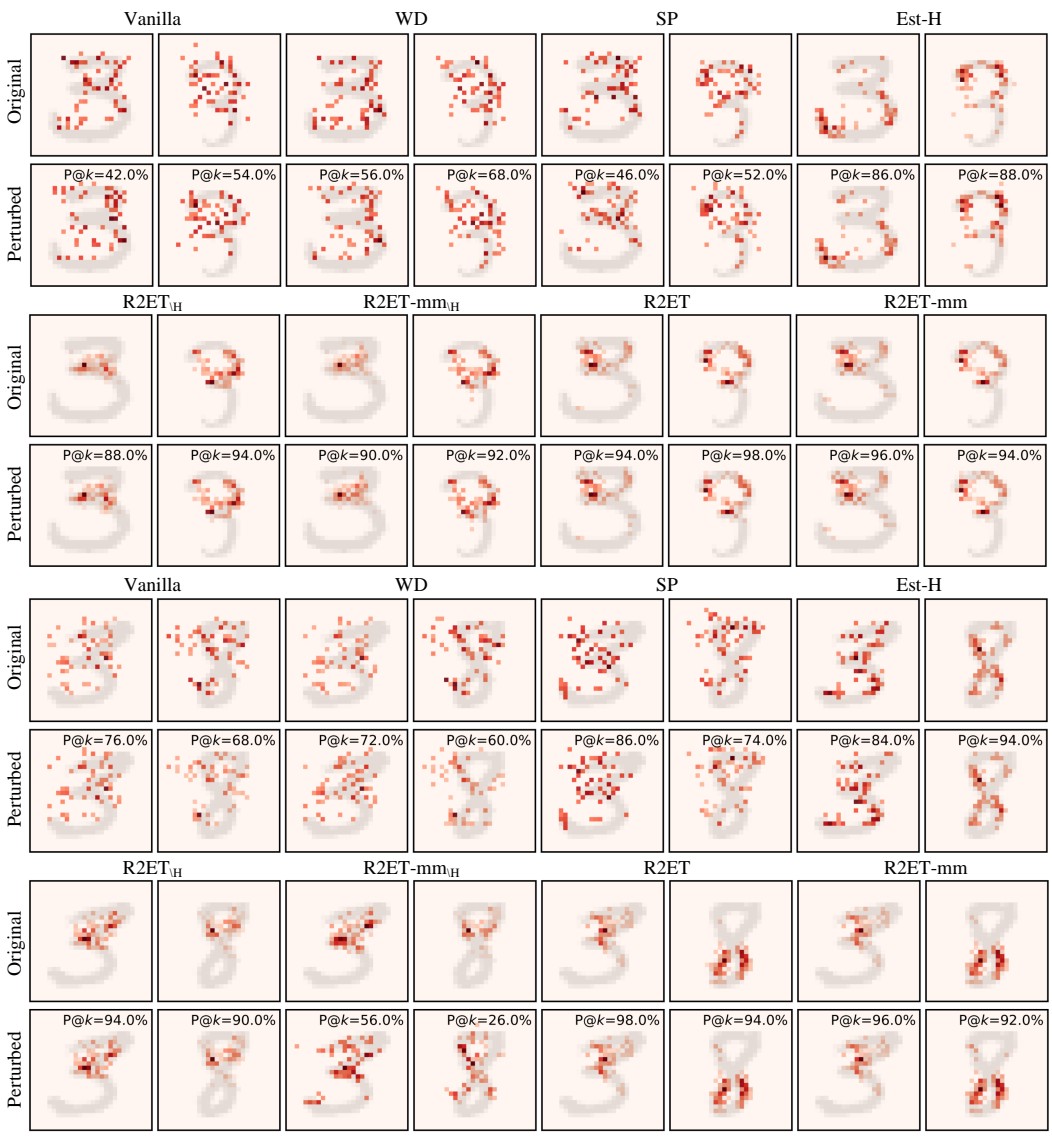

Figure 5: Saliency maps concerning the original image pair and the image pair perturbed under ERAttack for all methods. The red pixels are the top 50 important features in saliency maps, with darker colors meaning more important.

Scatter plots in Fig. 6 show different constrained attack methods in terms of P@$k$ and sensitivity. The attacker looks for lower sensitivity and P@$k$: lower sensitivity means that the constraints are better satisfied, and lower P@$k$ means that more top-$k$ important features in the explanation are distorted. The asterisks highlight the attack methods having the smallest P@$k$ and no more than 2% sensitivity. These attack methods are used to evaluate the defense strategies in Table 1. We do not use constrained attacks on three tabular datasets and MNIST because their sensitivity is 0%. Besides, Table 6 shows that the difference between cAUC and aAUC for all methods is no more than 0.01, indicating the success of the selected constrained attack to retain the predictions.

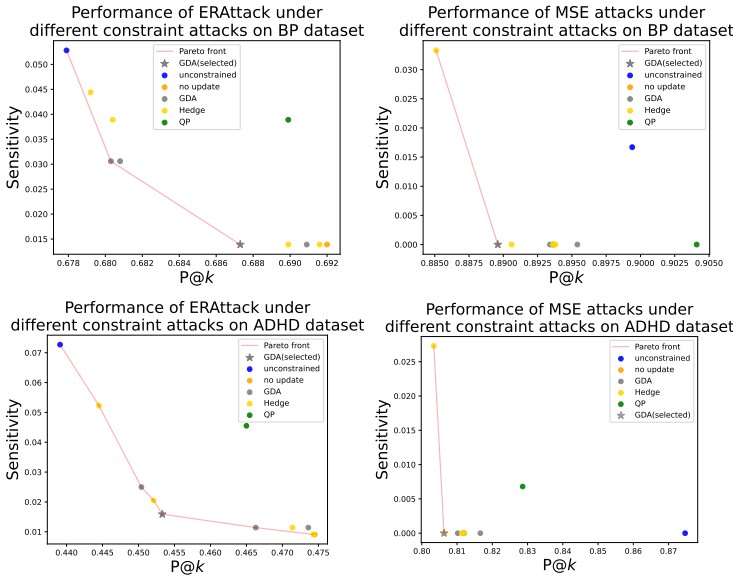

Figure 6: Performance of different constrained ERAttack and MSE attacks on BP and ADHD for Vanilla. Points in different colors represent different constraint attack methods, and the same color represents the same method with different step sizes. The red line implies the Pareto front, and the asterisk marks the selected attack method.

Table 6: **cAUC/aAUC** of SNs trained by different methods under ERAttack and MSE attack on ADHD and BP.

| Method | ADHD(ERAttack) | BP(ERAttack) | ADHD(MSE attack) | BP(MSE attack) |
|---|---|---|---|---|
| Vanilla | 0.7663 / 0.7729 | 0.6812 / 0.6920 | 0.7663 / 0.7659 | 0.6744 / 0.6776 |
| WD | 0.7513 / 0.7582 | 0.6739 / 0.6726 | 0.7508 / 0.7542 | 0.6753 / 0.6722 |
| SP | 0.7443 / 0.7358 | 0.6767 / 0.6817 | 0.7395 / 0.7375 | 0.6706 / 0.6763 |
| Est-H | 0.7619 / 0.7643 | 0.6576 / 0.6594 | 0.7618 / 0.7633 | 0.6572 / 0.6558 |
| AT | 0.7325 / 0.7277 | 0.6405 / 0.6347 | 0.7649 / 0.7659 | 0.6728 / 0.6773 |
| R2ET$_{\backslash H}$ | 0.7090 / 0.7153 | 0.6697 / 0.6732 | 0.7061  0.7058 | 0.6665 / 0.6700 |
| R2ET-mm$_{\backslash H}$ | 0.7099 / 0.7008 | 0.6711 / 0.6720 | 0.7020 / 0.7014 | 0.6642 / 0.6654 |
| R2ET | 0.7169 / 0.6973 | 0.6833 / 0.6839 | 0.7049 / 0.7104 | 0.6738 / 0.6780 |
| R2ET-mm | 0.7590 / 0.7633 | 0.6892 / 0.6957 | 0.7583 / 0.7580 | 0.6841 / 0.6852 |

### B.2.4 Trade-off between Accuracy and Explanation Robustness

A good defensive strategy should be both robust and accurate. Authors in [78] find the trade-off between *prediction* robustness and prediction performance. We explore the trade-off between the *explanation* robustness (P@$k$) and prediction performance (cAUC/aAUC). A defender looks for a higher P@$k$ and AUCs. In Fig. 7, at least one R2ET and its variants are on the Pareto front on all datasets, demonstrating that R2ET and its variants are more advantageous in the trade-off between robustness and accuracy. Furthermore, on MNIST, compared with other methods on the Pareto front,

R2ET variants on the Pareto front sacrifice less AUC (less than 2%) but gain significant improvements on P@$k$ (20% $\sim$ 40%). On BP, Est-H improves P@$k$ by 0.01 but loses about 4% AUC. R2ET on ADHD with both high AUC and the highest P@$k$ indicates the possibility that a model can be precise and explanation-robust at the same time.

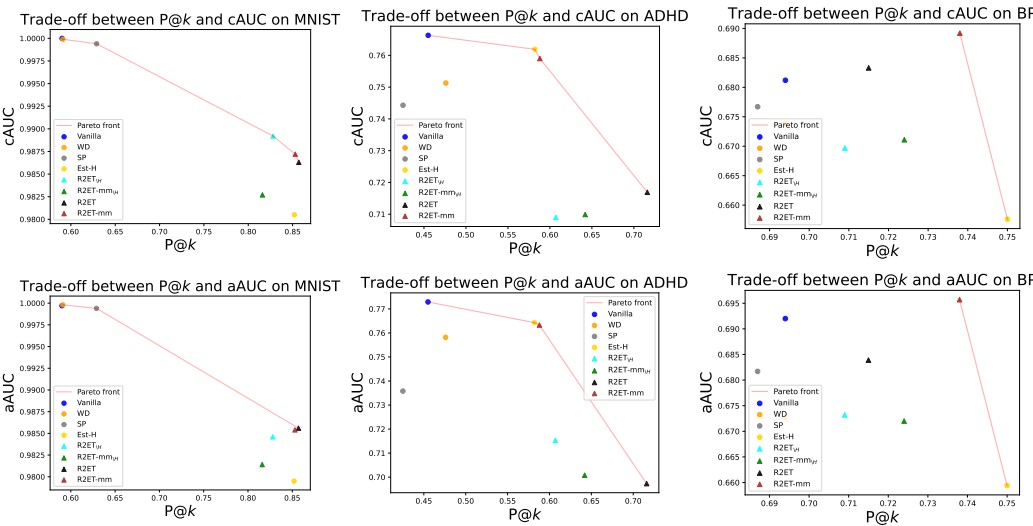

Figure 7: Trade-off between explanation robustness (P@$k$) and prediction performance (cAUC/aAUC) on MNIST, BP and ADHD. The red lines imply the Pareto front, and the triangles present R2ET and its variants.

### B.2.5 Sensitivity Analysis

Table 7: P@$k$ (shown in percentage) of models trained by different methods under ERAttack. Three numbers on tabular datasets present the results when $k$ is **2**, **5**, and **8**, respectively. $k$ is set to **10**, **30**, and **50** on the rest datasets, respectively.

| Method | Adult | Bank | COMPAS | MNIST | ADHD | BP |
|---|---|---|---|---|---|---|
| Vanilla | 79.3 / 81.9 / 87.6 | 97.9 / 81.4 / 83.0 | 66.1 / 78.1 / 84.2 | 51.6 / 56.7 / 59.4 | 39.4 / 43.2 / 45.3 | 63.8 / 67.3 / 68.7 |
| WD | **100.0** / 99.8 / 91.7 | 97.2 / 92.5 / 82.4 | **100.0** / 88.1 / 87.7 | 51.6 / 56.6 / 59.9 | 41.0 / 46.3 / 48.6 | 63.6 / 68.0 / 68.8 |
| SP | **100.0** / **100.0** / 97.4 | **100.0** / **99.4** / 95.4 | **100.0** / **100.0** / **99.5** | 54.9 / 60.5 / 63.1 | 39.4 / 42.7 / 44.7 | 63.9 / 66.9 / 67.9 |
| Est-H | **100.0** / 92.9 / 87.1 | 86.9 / 83.9 / 78.4 | 99.9 / 83.9 / 82.6 | 80.2 / 82.9 / 84.5 | 53.3 / 56.4 / 57.1 | **70.8** / **74.8** / 74.3 |
| Exact-H | **100.0** / 92.3 / 89.6 | 92.9 / 89.1 / 81.9 | 79.9 / 79.3 / 77.2 | - / - / - | - / - / - | - / - / - |
| SSR | **100.0** / 92.9 / 91.2 | 87.0 / 85.8 / 76.3 | 87.4 / 87.1 / 82.1 | - / - / - | - / - / - | - / - / - |
| R2ET$_{\backslash H}$ | **100.0** / 98.5 / **97.5** | **100.0** / 96.0 / **100.0** | 84.0 / 98.1 / 91.9 | 77.6 / 80.7 / 82.3 | 59.3 / 62.6 / 63.1 | 64.5 / 69.3 / 71.4 |
| R2ET-mm$_{\backslash H}$ | **100.0** / 99.8 / 93.5 | 99.2 / 94.8 / 95.8 | **100.0** / 83.7 / 95.3 | 76.5 / 80.0 / 81.9 | 57.2 / 62.3 / 64.3 | 67.6 / 71.5 / 73.1 |
| R2ET | **100.0** / 93.2 / 92.1 | **100.0** / 91.8 / 80.4 | 79.4 / 88.8 / 92.0 | **81.4** / **84.1** / **85.3** | **70.7** / **74.1** / **73.8** | 66.1 / 70.5 / 72.2 |
| R2ET-mm | **100.0** / 99.7 / 87.8 | 98.6 / 92.2 / 75.1 | 89.0 / 89.0 / 82.1 | 80.6 / 83.5 / 84.9 | 55.3 / 58.5 / 60.1 | 69.4 / 74.2 / **75.1** |

In this section, we consider the impacts of three hyperparameters or settings.

**Impacts of $k$.** Table 7 shows that R2ET and its variants stay at the top compared with other baselines for various $k$. Besides, we further explore the more fundamental reasons why models perform differently for various $k$. As mentioned in Eq. (5), the gaps of gradients of original inputs positively contribute to the thickness. In Fig. 8, we sort the gradients of Vanilla models in descending order where their gaps can be inferred. Vanilla achieves about 100% P@$k$ when $k = 2$ on Adult and Bank, indicating that even ERAttack cannot effectively manipulate the ranking in such scenarios. The reason is that the top 2 features are much more significant than the rest as shown in Fig. 8. However, there is a narrow margin between the top 2 and top 3 features on COMPAS, and attackers can easily flip their relative rankings, thus Vanilla's P@$k$ reduces to around 2/3. On MNIST, BP and ADHD, Vanilla models' P@$k$ increases as $k$ grows. It seems that models are more efficient against ERAttack with larger $k$. However, the *absolute number* of success manipulations for top-$k$ features increases for larger $k$. Take Vanilla on MNIST as an example, ERAttack distorts the model's 5 features when $k$=10,

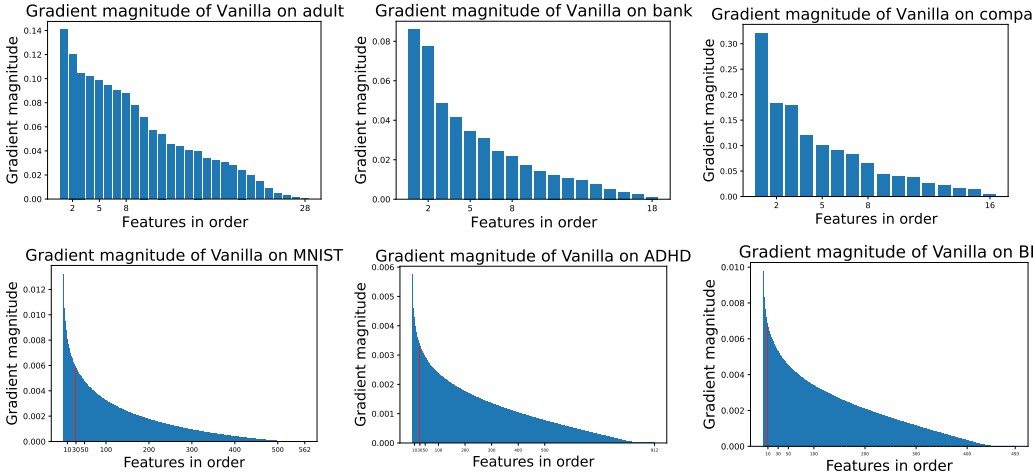

Figure 8: We order the features based on the *Vanilla* model's gradient magnitude with respect to each original input on different datasets. Notice that these figures would differ for various models.

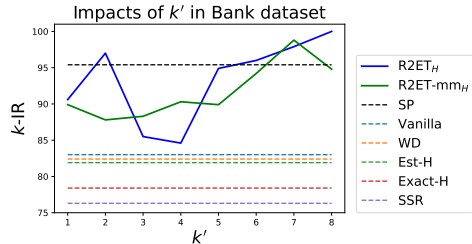

Figure 9: Sensitivity analysis on the number of selected pairs. R2ET$_{\backslash H}$ and R2ET-mm$_{\backslash H}$ are examined in Bank dataset.

and about $40\% * 50$=20 features when $k$=50. Since ERAttack uses the same budget for different $k$, it becomes harder for ERAttack to manipulate more features simultaneously (for larger $k$). However, it indeed kicks off more features from the top positions due to narrower margins on the long tail.

**Impacts of the number of selected pairs $k'$.** As discussed in Sec. 4.2, fewer ($k'$) compared feature pairs for R2ET and its variants alleviate optimization issues. Specifically, R2ET takes $\sum_{i=k-k'+1}^{k-1} h(\mathbf{x}, i, k) + \sum_{j=k+1}^{k+k'+1} h(\mathbf{x}, k, j)$ as objective. Since R2ET$_{\backslash H}$ and R2ET-mm$_{\backslash H}$ are the two best variants in tabular datasets, we conduct a sensitivity analysis of these two methods on Bank. We set $k = 8$, and change $k'$ from 1 to $k$. Fig. 9 indicates that R2ET$_{\backslash H}$ and R2ET-mm$_{\backslash H}$ perform much better when $k' > 4$ than those with $k' \leq 4$. Apparently, when more pairs of features are taken into consideration, R2ET and its variants have a more comprehensive view of feature rankings to maintain the rankings better. Besides that, R2ET$_{\backslash H}$ and R2ET-mm$_{\backslash H}$ outperform almost all baselines, except for SP, for all $k'$.

**Impacts of pretrain / retrain.** Lastly, we explore how good these methods are when applying them in the retrain schema, and each model is retrained from the Vanilla model for at most 10 epochs. Since the Vanilla model has already converged and reached a good AUC, we assume that the Vanilla model's explanation ranking is an excellent reference. Thus these robust methods try to maintain the Vanilla model's rankings. The retraining will be terminated if P@$k$ between Vanilla and retrain models' rankings significantly drops, or the retrain model's cAUC drops a lot. Table 8 presents the results for comparing two training schemas. Since SP changes the models' structure (activation function), we do not consider it here. Instead, we add one more baseline, CL [28], because retraining demands much fewer training epochs, thus its time complexity is acceptable. More details for CL can be found in Sec. 4.2. Besides that, *none* of retrain models by Exact-H and SSR can maintain Vanilla model's explanation rankings and cAUC at the same time in Bank dataset, and thus both are not applicable.

Table 8: P@$k$ (shown in percentage) of models trained by different methods under ERAttack ($k = 8$). Two numbers indicate the performance of models being **trained from scratch** or **retrained from Vanilla models**.

| Method | Adult | Bank | COMPAS |
|---|---|---|---|
| Vanilla | 87.6 / 87.6 | 83.0 / 83.0 | 84.2 / 84.2 |
| WD | 91.7 / 88.3 | 82.4 / 82.1 | 87.7 / 82.7 |
| CL | - / 93.1 | - / **100.0** | - / 87.1 |
| Est-H | 87.1 / 92.1 | 78.4 / 85.2 | 82.6 / 85.1 |
| Exact-H | 89.6 / 88.7 | 81.9 / - | 77.2 / 87.0 |
| SSR | 91.2 / 88.7 | 76.3 / - | 82.1 / 86.1 |
| R2ET$_{\backslash H}$ | **97.5 / 100.0** | 100.0 / 100.0 | 91.9 / **97.8** |
| R2ET-mm$_{\backslash H}$ | 93.5 / 100.0 | 95.8 / 98.3 | **95.3** / 95.6 |
| R2ET | 92.1 / 92.6 | 80.4 / 86.2 | 92.0 / 85.1 |
| R2ET-mm | 87.8 / 91.6 | 75.1 / 86.2 | 82.1 / 87.4 |

### B.2.6 Faithfulness of explanations on different models

We report the faithfulness of explanations evaluated by three widely used metrics, DFFOT, COMP and SUFF, in Table 9.

Table 9: Faithfulness of explanations evaluated by DFFOT ($\downarrow$) / COMP ($\uparrow$) / SUFF ($\downarrow$).

| Method | Adult | Bank | COMPAS | MNIST | ADHD | BP |
|---|---|---|---|---|---|---|
| Vanilla | 0.24 / 0.43 / 0.18 | 0.23 / 0.14 / 0.04 | **0.17** / 0.37 / 0.14 | 0.37 / 0.16 / 0.23 | 0.51 / 0.05 / 0.28 | 0.40 / 0.06 / 0.29 |
| WD | 0.45 / 0.47 / 0.23 | 0.36 / 0.27 / 0.07 | 0.29 / 0.41 / 0.18 | 0.37 / 0.16 / 0.22 | 0.49 / 0.06 / 0.27 | **0.35** / 0.05 / 0.33 |
| SP | 0.43 / 0.47 / 0.25 | 0.35 / 0.31 / 0.07 | 0.29 / **0.45** / 0.18 | 0.38 / 0.15 / 0.22 | **0.30** / 0.10 / 0.34 | 0.38 / **0.06** / 0.30 |
| Est-H | 0.44 / 0.44 / 0.24 | 0.18 / 0.21 / 0.06 | 0.27 / 0.42 / 0.17 | 0.23 / 0.24 / **0.18** | 0.59 / 0.04 / 0.26 | 0.45 / 0.05 / **0.24** |
| Exact-H | 0.43 / 0.46 / 0.23 | 0.19 / 0.14 / 0.04 | 0.30 / 0.40 / 0.18 | - / - / - | - / - / - | - / - / - |
| SSR | 0.54 / 0.39 / 0.21 | 0.46 / 0.04 / **0.01** | 0.32 / 0.43 / 0.18 | - / - / - | - / - / - | - / - / - |
| AT | 0.16 / 0.14 / **0.08** | 0.19 / 0.10 / 0.03 | 0.24 / 0.10 / **0.07** | 0.40 / 0.12 / 0.28 | 0.35 / 0.10 / 0.26 | 0.46 / 0.06 / 0.25 |
| R2ET$_{\backslash H}$ | **0.13** / **0.50** / 0.14 | 0.34 / 0.32 / 0.10 | **0.17** / 0.40 / 0.17 | 0.23 / 0.22 / 0.19 | 0.38 / 0.13 / 0.37 | 0.43 / **0.07** / 0.29 |
| R2ET-mm$_{\backslash H}$ | 0.42 / 0.47 / 0.22 | 0.34 / **0.41** / 0.14 | 0.25 / 0.42 / 0.17 | 0.25 / 0.22 / 0.21 | 0.37 / **0.17** / 0.37 | 0.42 / **0.07** / 0.29 |
| R2ET | 0.32 / 0.46 / 0.19 | **0.11** / 0.24 / 0.07 | 0.27 / 0.39 / 0.17 | **0.18** / **0.26** / 0.23 | 0.48 / 0.12 / 0.26 | 0.42 / **0.07** / 0.29 |
| R2ET-mm | 0.38 / 0.48 / 0.20 | 0.12 / 0.21 / 0.08 | 0.28 / 0.44 / 0.15 | 0.19 / **0.26** / 0.22 | 0.50 / 0.04 / **0.25** | 0.45 / 0.05 / 0.29 |

## C Related Work

**Explainable machine learning and explanation robustness.** Recent post-hoc explanation methods for deep networks can be categorized into gradient-based [103, 71, 7, 75, 77, 73, 100, 98, 35], surrogate model based [64, 30, 82], Shapley values [47, 12, 45, 99, 3], and causality [58, 10, 56]. The gradient-based methods are widely used in practice due to their simplicity and efficiency [55], while they are found to lack robustness against small perturbations [23, 29]. Some works [13, 74, 32, 83, 70, 80] propose to improve the explanation robustness by time-consuming adversarial training (AT). To bypass the high time complexity of AT, some works propose replacing ReLU function with softplus [17], training with weight decay [16], and incorporating gradient- and Hessian-related terms as regularizers [17, 88, 91]. Some works propose new *explanation* methods, rather than *training* methods, to enhance explanation robustness [46, 44, 11, 51, 65, 75]. Besides, many works [50, 79, 87, 52, 67, 63, 97, 28, 96, 14, 78, 90, 101, 18] for improving adversarial robustness focus on *prediction* robustness, instead of *explanation* robustness, and thus are different from our work. The work [97] proposes the *prediction* thickness by measuring the distance between two decision boundaries, while our explanation thickness qualifies the expected gaps between two features' importance. Furthermore, the explanation thickness is inherently more complicated to optimize due to the nature of gradient-based explanations being defined as first-order derivative relevant terms.

**Ranking robustness in IR.** Ranking robustness in information retrieval (IR) concerning noise [106] and adversarial attacks [26, 104, 105, 42, 85] is well-studied. Ranking manipulations in IR and explanations are different because 1) In IR, authors either manipulate the position of *one* candidate [104, 26], or manipulate a query to distort the ranking of candidates [104, 105, 42]. We

manipulate input to swap *any* pairs of salient and non-salient features. 2) Ranking in IR is based on the model predictions. However, explanations are defined by gradient or its variants, and studying their robustness requires second or higher-order derivatives, and it motivates us to design an efficient regularizer to bypass costly computations.

**Top-$k$ intersection.** Top-$k$ intersection is widely used to evaluate explanation robustness [13, 88, 74, 32, 83, 70, 23]. However, most existing works study the explanation robustness through $\ell_p$ norm [13, 16, 17, 88], cosine similarity [83, 84], Pearson correlation [32], Kendall tau [83], or KL-divergence [70]. These correlation metrics measure the explanation similarity using *all* features, and a high similarity can have entirely different top salient features. To optimize the top-$k$ intersection in *classification* tasks, some works propose various surrogate losses based on the upper bounds of top-$k$ intersection [2, 37, 38, 39, 8, 95, 60]. The upper bounds hold for predictions whose gaps are not larger than one, while it is not always true for gradient-based explanations with unbounded values, which prevents adopting similar methods to *explanation* tasks.

**Distributional shift.** The distributional shifts [61, 41] and our work are distinct concerning the "perturbation budget" between the original and perturbed inputs: In contrast, the distributional shifts ensure reasonable predictions (and explanations) for out-of-distribution samples, where a substantial deviation from the in-distribution samples is expected, naturally leading to different explanations and rankings. The input perturbations are intentionally neglectable in ours, and other adversarial robustness works, for stealthiness.

