# OpenReview forum: "Training for Stable Explanation for Free"
_NeurIPS.cc/2024/Conference — NeurIPS 2024 poster_

### Official Review · Reviewer_4WDh · 2024-07-11

**Soundness:** 4
**Presentation:** 3
**Contribution:** 3
**Rating:** 5
**Confidence:** 4

**Summary:**

The paper proposes a novel metric for assessing the stability of explanations in machine learning models, which is crucial for their trustworthiness. The authors introduce a method called R2ET (Robust Ranking Explanation via Thickness) designed to train models to generate stable explanations efficiently and effectively. They provide experiments across various data modalities and model architectures, showing that R2ET achieves superior stability against stealthy attacks and generalizes effectively across different explanation methods.

**Strengths:**

1. The introduction of a new metric that aligns more closely with human perception than existing $\ell_p$ distance measures is compelling.
2. The theoretical grounding of the methods and the extensive empirical validation provide a high degree of confidence in the results.
3. The paper is clearly written.

**Weaknesses:**

1. The paper’s discussion of explanation robustness is focused on adversarial robustness. However, the explanation robustness can also be affected by other factors, such as distributional shifts [1, 2]. It would be beneficial to discuss the relationship and difference between the proposed method and these methods.
2. While the paper tests various data modalities, the focus is small-scale datasets, such as MNIST. Could you discuss the potential limitations of the R2ET method when applied to real-world, large-scale datasets? An evaluation of the scalability of the proposed method is beneficial.

[1] Li et al. “Are Data-driven Explanations Robust against Out-of-distribution Data?” CVPR 2023.

[2] Pillai et al. “Consistent explanations by contrastive learning” CVPR 2022.

**Questions:**

Should the authors address the identified weaknesses (listed in descending order of importance), I would be inclined to raise my rating. These revisions would enhance the paper’s contribution to the field and its practical applicability.

**Limitations:**

The discussion of limitations in the paper could be more comprehensive. While the authors mention general applicability issues, they do not delve into specific limitations that might affect the deployment of their method in real-world settings, particularly in scenarios with highly variable and noisy data.

---

> ### Author Rebuttal · Authors · 2024-08-07
>
> Thanks for your detailed and constructive reviews.
>
> > To Weakness 1: The paper’s discussion of explanation robustness is focused on adversarial robustness. However, the explanation robustness can also be affected by other factors, such as distributional shifts [1, 2]. It would be beneficial to discuss the relationship and difference between the proposed method and these methods.
>
> 1. We believe that adversarial robustness (AR) and distributional shifts (DS) as highlighted in [1,2] are **distinct** because of the “perturbation budget” $\epsilon$ between the original and perturbed inputs: $\epsilon$ is **intentionally small** in AR for stealthiness, but **relatively large** in DS due to the more substantial variations inherent in out-of-distribution (OOD) samples, making DS and AR **incomparable**.
>
> Specifically, the goal of AR is to **maintain the explanations unchanged** even with *neglectable* input perturbations, aiming for consistency in explanations. In contrast, DS ensures **reasonable predictions (and explanations) for OOD samples**, where a *substantial deviation* from the in-distribution samples is expected, naturally leading to **different explanations and rankings**.
>
> For example, two images of birds (e.g., Figure 2 from [1]), although from the same class, differ significantly in shape and location, leading to varied feature importance across the images. This variability is typical in DS scenarios. Conversely, in AR cases, the visual and explanatory differences are minimal, thereby requiring the explanation rankings to remain consistent.
>
> 2. Despite the fundamental differences of $\epsilon$ between AR and DS, the concepts and properties of thickness and R2ET, as detailed in Section 4, are not specified to AR, but **broadly applicable to different robustness scenarios**. Thus, we can view them in the DS frameworks:
>
> Specifically, the variable $\mathcal{D}$ in Definition 4.1 and 4.2 (definition of thickness) denotes the (unknown) distribution of sample $x'$. The thickness can be defined under DS framework by viewing $x\in\mathcal{X}$ as in-distribution samples, and $x'\in\mathcal{D}$ as out-of-distribution samples. Furthermore, Propositions 4.4 – 4.7 do not rely on specific assumptions about the distribution $\mathcal{D}$ and merely limit the perturbation budget to $\epsilon$. Thus, these propositions hold even if $\mathcal{D}$ represents the out-of-distributions. This adaptability indicates the universal applicability of thickness and R2ET, making them address both AR and DS challenges effectively.
>
> > To Weakness 2: While the paper tests various data modalities, the focus is small-scale datasets, such as MNIST. Could you discuss the potential limitations of the R2ET method when applied to real-world, large-scale datasets? An evaluation of the scalability of the proposed method is beneficial.
>
> We want to point out that we also include ROCT image dataset from *real-world medical applications* (where robust explanation is of critical importance) in our experiments. The dataset consists of images having 771x514 pixels on average, making it **comparable in scale** to well-known datasets such as ImageNet (469x387), MS COCO (640x480), and CelebA (178x218).
>
> Regarding the scalability of the R2ET method, we analyze the computational efficiency of R2ET, highlighting that it requires **only two additional backward propagations**. This makes R2ET considerably more efficient (**20x faster**) than many adversarial training (AT) methods, as well as [1] and [2]. More discussions are disclosed in Lines 810-818. This efficiency suggests that R2ET’s approach to generating robust explanations is scalable to larger, more complex datasets while maintaining computational feasibility.
>
>
> > To Limitation: The discussion of limitations in the paper could be more comprehensive. While the authors mention general applicability issues, they do not delve into specific limitations that might affect the deployment of their method in real-world settings, particularly in scenarios with highly variable and noisy data.
>
> We recognize that the limitations discussed could be expanded to better address scenarios that involve highly variable and noisy data, which fall outside the “small perturbation” scenarios. While the theoretical analyses in Sections 4-5 will still hold across varying distributions and perturbations, affirming that our approach is theoretically sound under unknown perturbations. It points to a valuable direction for future research: testing and potentially adjusting R2ET to ensure robustness and reliability in more diverse and challenging environments.

---

> > ### Comment · Reviewer_4WDh · 2024-08-13
> > **Thank you for the reply**
> >
> > Thank you for your reply; I am happy to raise my score.

---

> > > ### Author Response · Authors · 2024-08-13
> > >
> > > Thank you so much!

---

### Official Review · Reviewer_VSmU · 2024-07-12

**Soundness:** 3
**Presentation:** 3
**Contribution:** 3
**Rating:** 7
**Confidence:** 4

**Summary:**

The paper aims at a robust explanation of predictive models. A new concept called “feature ranking robustness” is proposed and a corresponding objective function called “explanation thickness” and an optimization algorithm R2ET is designed to increase the thickness during model training time. Theoretical analyses that covers the numerical and statistical aspects of the thickness are provided. Experiments results on diverse data modalities including tabular, image, and graph data, demonstrate the robustness of the explanations obtained from the thickness metric.

**Strengths:**

1) The enhancement of explanation robustness is different from prediction robustness as higher-order gradients are involved. The submission consider a new metric based on relative ranking of top-k salient features rather than distance between saliency maps over all features, thus focusing more on the important features that will be perceived by the users.
2) he theoretical analyses are relevant and novel. In particular, the authors connect thickness to certified robustness, adversarial training, and constrained optimization (section 4.2), and the multi-objective optimization analysis help clarify the trade-off between model accuracy and explanation robustness. The experimental results are solid, with many datasets, baselines, and metrics to verify the proposed method.
3) The paper is well-organized and concepts are clearly defined.
4) while explanation robustness has been studied previously, the paper adopts a new perspective of top-k thickness. Then this concept results in a new optimization algorithm and in-depth theory. In particular, formulating explanation problem under the MOO framework and the learning theoretical framework make unique and significant contribution to the XAI community. It seems that the R2ET algorithm can be applied to multiple different gradient-based explanation methods.

**Weaknesses:**

-	The thickness concept has been used for prediction robustness.
-	Eq. (6) involves evaluating the Hessian matrix and that can be quite time consuming, compared to WD that does not need the Hessian matrix.

**Questions:**

-	In Figure 2, why the correlation indicates that thickness is a better metric of explanation robustness? More details are needed.
-	When applying R2ET to other explanation methods, do you need to modify the R2ET algorithm? In other words, how easy it is to apply R2ET to a broader spectrum of explanation methods?

---

> ### Author Rebuttal · Authors · 2024-08-07
>
> Thanks for your detailed and constructive reviews.
>
> >To Weakness 1: The thickness concept has been used for prediction robustness.
>
> Their fundamental meanings differ: the thickness for prediction robustness measures the distance between two decision boundaries, while explanation thickness qualifies the expected gaps between two features' importance. Furthermore, different from the “boundary thickness” for prediction robustness, our explanation thickness is inherently more complicated to optimize due to the nature of gradient-based explanations being defined as first-order derivative relevant terms. Thus, we try to speed up the computation and avoid adversarial training.
>
> > To Weakness 2: Eq. (6) involves evaluating the Hessian matrix and that can be quite time consuming, compared to WD that does not need the Hessian matrix.
>
> While Eq. (6) requires calculating the Hessian norm, we address the potential computation burden by the finite difference. It allows us to **estimate the Hessian norm effectively with only two additional backward propagations**. Thus, compared to Vanilla and WD, R2ET remains competitive in terms of computations. R2ET is much more time-saving comparing many existing baselines, such as about **20x faster** than adversarial training. More discussions concerning time complexity are in Lines 810-818.
>
>
> > To Question 1: In Figure 2, why the correlation indicates that thickness is a better metric of explanation robustness? More details are needed.
>
> The left subfigure demonstrates a high correlation between the sample’s thickness and the attacker’s required budget to manipulate the rankings. This high correlation signifies that samples with greater thickness demand a larger attack budget for a successful manipulation, thereby justifying that thickness serves as a metric for evaluating explanation robustness.
>
> Conversely, the right subfigure shows a low correlation between the Hessian norm of samples and the required attack budget. It indicates that the difficulty of manipulating explanations remains similar across samples, regardless of whether they have a large or small Hessian norm. This low correlation suggests that the Hessian norm is not a reliable indicator of explanation robustness.
>
> > To Question 2: When applying R2ET to other explanation methods, do you need to modify the R2ET algorithm? In other words, how easy it is to apply R2ET to a broader spectrum of explanation methods?
>
> R2ET is general to *a range of gradient-based explanation methods*—including Gradient (Grad), Gradient*Input, SmoothGrad, and Integrated Gradients (IG)—**without any modifications** given theoretical analysis (see Appendix A.1.2) and experimental outcomes (Section 6).
>
> In the future, we aim to extend R2ET to a broader spectrum of explanation methods *beyond gradient-based ones*. Given the objective of R2ET that directly maximizes the gaps between feature importances and minimizes the Hessian norm, R2ET should also perform effectively with other explanation methods.

---

> ### Comment · Reviewer_VSmU · 2024-08-13
>
> I will keep my score after reviewing other referees' comments and the manuscript.

---

> > ### Author Response · Authors · 2024-08-14
> >
> > Thank you once again. We truly appreciate your insightful and valuable feedback on our work.

---

### Official Review · Reviewer_9MCa · 2024-07-15

**Soundness:** 3
**Presentation:** 2
**Contribution:** 2
**Rating:** 6
**Confidence:** 2

**Summary:**

The paper describes a regularizer to add to a loss function to encourage the resulting model to
have input attributions robust to ranking changes in its top k features. That is, for an input x,
the input attributions (a score for each input) for this input will be similarly ordered (at least
in the top k scoring features) as compared to the input attribution of any small perturbation x' of
x. The argument to use ranking and top-k ranking for defining robustness is that only the top few
input features are of relevance to a human interpreting an explanation. Ranking robustness itself
might be because the relative importance of features is more relevant to the human than the
absolute magnitudes of the importance.

To make ranking robust, the regularizer encourages gaps or "thickness" in the scores of top-k most
important features which would naturally make it harder to reorder them via a small perturbations
in the inputs and therefore (by addition of Hessian to encourage smoothness of gradients) feature
importance scores (which are assumed to be gradients or "saliency"). Analytical arguments are made
as to how to effectively implement the regularizer in a more efficient fashion. Analytical
arguments are presented as to a worst-case performance of an attack on the proposed method (though
why worst case is relevant confuses me).

A large collection of experiments demonstrate the use of the regularizer (termed R2ET) with various
parameters and compared to other robustness methods faced with several adversarial explanation
manipulation methods. These results show the proposed method is not optimal in all cases though it
is in many.

**Strengths:**

+ S1. The approach includes analytical and experimental methods and seems to achieve to some extent
  the ranking robustness goal.

+ S2. Multiple adversarial attacks experimented with including one aimed at ranking specifically.

+ S3. Multiple other robustness methods experimented with including several that approximate some
  of the same loss terms (i.e. the Hessian) in different ways (though lack of experimental timing
  information is a shortcoming as listed below).

**Weaknesses:**

- W1. Motivation is based on ranking robustness but ranking robustness is not directly measured in
  the experimental sections. In the collection of explanation metrics enumerates in B.1.4, the P@k
  only partially exposes robustness with respect to ranking thought would miss all importance swaps
  that happen within the top k features. The experimental discussion seems to be making the case
  that the proposed top-k thickness metric is related to P@K but as P@k does not fully capture
  ranking robustness, those arguments also fail to make the connection. While I agree that
  optimizing for thickness would promote ranking stability, I do not find the concepts identical
  (i.e. there may be other or better ways to achieve ranking stability). Some experimental results
  may be hinting at the mismatch where I see the regularizer without Hessian sometimes
  outperforming the one with it.

  Suggestion: Include a more direct ranking robustness measure in the experimental results. I'm
  unsure what form of such a measure would be. Perhaps ranking correlation from [103] can be used
  for feature importance in the present paper.

- W2. Computational effort reduction not demonstrated. Part of the design seems to be limit
  backwards passes or other computationally costly steps. The experiments are plenty but none seem
  to demonstrate this aspect. There is an analytical time complexity discussion in the "Time
  complexity of R2ET" paragraph. The title itself suggests that stability is free but is a bit
  deceptive in that the training incurs additional costs.

  Suggestion: Include the experimental measurements of effort in the results. One of the strengths
  of the present approach is that the explanation-time is less costly as saliency is relatively
  less costly than alternatives. Showing this would be good to discuss/experiment with as well.

- W3. Most (or all) theoretical analysis applies only to saliency. Arguments/discussions regarding
  Proposition 4.6 in the appendix include the assumption that the explanation is a gradient (a
  saliency explanation). Saliency are defined in Preliminaries but it is not clearly stated there
  that the rest of the paper assumes all explanations are of this form. As the methods target
  saliency, a lot of this paper depends on saliency being a good explanation method to begin with.
  However, plenty of works argue that it is not (see [73, 75]). Some benefits of things like IG
  also share motivation to ranking robustness (that relative scores are more important than
  absolute).

  Suggestion: Make it clear that the analytical sections make this assumption. Also, note in the
  "Apply R2ET to other explanation methods" paragraph in Section 6 that the analytical conclusions
  do not apply to these other methods. A good inclusion to the limitations discussion would be that
  saliency itself has some problems which may warrant use of a different explanation method
  especially for problems with regards to robustness which this work is attempting to solve using a
  regularizer.

Other comments/suggestions:

- C1. The intro/motivation could use some discussion of how the primary desiderata in explanations
  and how they related to each other. The paper is primarily based on ranking robustness which I
  could say is an element of an explanation's interpretability and resistance to adversarial
  manipulation. From here it would be useful to say that addressing robustness could be done with
  adjustments to the explanation method or to adjustment to the model or training. The paper
  discusses the second but a note about the benefits/drawbacks of the first would be good to
  include as they introduce some of the subsequent experiments which are presently hard to fit into
  the big picture.

  1. Want (interpretability and) resistance to adversarial manipulation.

  2. Option 1: Change the explanation method (don't use Saliency). Benefit: no model changes.
     Drawbacks: faithfulness vs robustness tradeoff. Explanation-time costs might be large.

     Option 2: Add regularizer for saliency during training. Benefits: Fast (or "free" as per
     title) explanations, better faithfulness, Drawbacks: training-time costs, utility loss.

     Option n: ???

  3. We do Option 2 and here are analytical and experimental results demonstrating the strengths
     and the impact of weaknesses.

  This chain of thought justifies the present (i.e. faithfulness) and suggested experiments (i.e.
  costs as per Weakness W2). Some of the experiments in the Appendix might be approaching these
  points though I'm having trouble connecting the dots.

- C2. The "thickness" notion could be better presented in the experiments. Figure 8 in the appendix
  shows some saliency/importance graphs without regularization for gaps. Can you include similar
  pictures for regularized models? I would expect to see the top-k features to have a linear
  decrease in importance. Seeing this (or something else) could go a long way in building intuition
  in the approach described in this paper.

Smaller things:

- The intro point about "inherent limitations of human cognition" as an argument for top-k
  importance doesn't apply as well to vision tasks where our eyes look at a huge number of raw
  features and necessarily so in order to resolve higher-level structures.

- The top-k thickness paragraph on line 97 cites [107] as a work suggesting maintaining ranking of
  features but this work is not on feature importance. The "raking" there is because the topic is
  ranking models.

- The statement "only the top-k important features in \cal{I}(x) are more relevant to human
  perception" is quite vague and I'm unsure how it could be formalized. Also note the grammar issue
  ("more relevant" than what?).

- "cost computations" -> "costly computations".

- "single-input DNNs", this reads like the DNNs have a single input neuron but doesn't mean that.
  Rephrase?

**Questions:**

- Q1. My understanding of a lot of the technical discussions in the paper was limited so the
  comments in the weaknesses in the prior section might not be fully justified. If this is the
  case, please address how the paper does not have the stated weakness. For those that are
  weaknesses, please comment on plausibility of addressing them either with my suggestions or
  otherwise.

- Q2. Proposition 4.6 relates Equation 6 and Equation 7. 7 includes perturbations while 6 does not.
  I'm having trouble qualifying the perturbations to make the statement make sense. Please include
  some higher-level points as to why this and other \theorems follow.

- Q3. Some results show that the regularization without the Hessian performs better than with it
  which is casting some doubt on the theoretical arguments. Alternatively it could be a mismatch
  between P@k and the optimization goal. Can ranking consistency be measured more directly? (i.e.
  as per suggestion of W1) and would it be able to explain the \H results?

- Q4. How would directional scores affect the methodology? While saliency has a sign, I don't see
  it demonstrated in this paper with a sign. Have all attributions been absolute valued first?
  Generally an explanation with both positive attribution and negative attribution would be more
  valuable and perhaps relative to ranking, having top-k (positive) and bottom-k (negative) would
  be the equivalent.

- Q5. I'm not following the purpose of worst-case complexity of an adversarial attack (Section 5).
  Wouldn't it make more sense to lower bound the complexity of an attack instead of upper bound it?

**Limitations:**

Only one limitation is noted in Section 7 with regards to using a surrogate of thickness. I believe
there are further limitations as noted in W3 for example. Societal impacts not mentioned though
addressing explanation robustness for its own sake or against adversarial perturbations is expected
to have only positive societal impact.

---

> ### Author Rebuttal · Authors · 2024-08-07
>
> Thanks for your detailed and constructive reviews.
> Due to the character limitation of rebuttal, we have to make the response concise and simple. Feel free to discuss them if more questions/concerns.
>
> > To W1:
>
> We agree that P@k may not be the “best” and “direct” metric for capturing ranking stability, and believe that the study of “best” evaluation measurement may be explored in future work, which potentially calls for psychology experts or human-involved surveys.
>
> We evaluate the explanation ranking robustness by P@k because 1) P@k is widely used to evaluate the explanation robustness in existing works [13, 86, 72, 32, 81, 68, 23]. 2) P@k better aligns with human perception. As shown in Figure 1, only the top few features (without their importance score nor relative rankings) will be delivered to end-users in credit card applications.
>
> Furthermore, as discussed in Lines 997-1000, while there are various metrics for ranking similarity—such as cosine similarity, Pearson correlation, and the one in [103]—these can be sensitive to changes in the ordering of less important features, which might not reflect human perception accurately as well. Because they often consider the entire rankings, potentially distorting their usefulness in scenarios where only the most critical features are relevant to decision-making.
>
> > To W2:
>
> While we include an analytical discussion on the time complexity of R2ET in the Appendix, we acknowledge that we did not present empirical data comparing the actual training times. Because the training time is influenced by several factors such as *convergence rates, hardware specifications, and learning rates*. Rather, we compare R2ET with baselines using **similar analytical training resources** (number of backward propagations), except Exact-H and SSR which are much more costly. Besides, we agree that demonstrating the computational efficiency of R2ET during explanation time could significantly strengthen our argument, as R2ET adds no cost to the existing gradient-based explanation methods.
>
> > To W3:
>
> In the revised version, we will clarify that the theoretical analysis applies to gradient (saliency) explanations, as initially introduced in the preliminary. Although the primary results are on saliency explanations, we extend our work by showing that R2ET could generalize to other mentioned explanation methods, such as Grad*Inp, IG and SmoothGrad, by maximizing their lower bounds (Appendix A.1.2). The experimental results further indicate R2ET could fit more explanation methods.
>
> > To C1:
>
> Thanks a lot. We agree with your comments, and will partially modify our writings logics in the revised version according to your suggestions.
>
> > To C2:
>
> We report “thickness” in Figure 2, Table 3, and Table 5, to show the strong correlation between thickness and robustness. The gaps shown in Figure 8 are not exactly the same as the thickness. We will include more figures for R2ET, and it is supposed to see a significant gap between the $k$-th and $(k+1)$-th features, aligning with R2ET's objectives.
>
> > To "small things":
>
> For small thing 1: We agree that humans may resolve higher-level structures and discuss such high-resolution images in the “concept-based” explanation cases, where top-k *sub-regions* may be of more importance than others.
> For others, we will modify our writing and make them clearer.
>
> > To Q1:
>
> Please refer to the response above.
>
> > To Q2:
>
> The connection is the perturbation budget $\epsilon$. Intuitively, Eq. (5) shows that a L-locally Lipschitz model’s local behavior under an $\epsilon$ manipulation can be approximated using $\epsilon$ and its high-order derivative information (such as Hessian). A large $\epsilon$ (for a stronger manipulation) leads to a rigorous restriction to the model (e.g., smaller change rates). Based on Eq. (5), R2ET in Eq. (6) “integrates” $\epsilon$ into $\lambda_2$. On the other hand, Eq. (7) aligns with two objective functions with $\epsilon$ as shown in Lines 629-634.
>
> > To Q3:
>
> We have justified the use of P@k in response to W1.
>
> As shown in Table 3, the model with the highest thickness leads to the best P@k. Thus, we attribute the super performance of R2ET\H to their higher thickness than R2ET. In other words, R2ET does not always achieve the best performance *may* because of the potential mismatch between **R2ET’s optimization goal** and **thickness**. Although we agree that there *may* be a mismatch between **P@k** and **“real ranking consistency”**. The later mismatch will be one of our future research directions.
>
> > To Q4:
>
> Our analysis and methods in Sections 4 and 5 are independent of the signs of gradients. In other words, the statements and proofs will hold in the general case where saliency does or does not have a sign. In the experiments, we follow existing works to take the magnitude (absolute values) of the gradient.
>
> > To Q5:
>
> We study the “worst-case” for attackers in Sec. 5, which provides a guarantee for a successful attack on the explanation rankings. In practice, the attacker could be limited to its attack (perturbation) budget and computation resources (number of attack iterations). Thus, the guaranteed successful attack is defined by the upper bound of the attack iterations.
>
> > To limitation:
>
> We will show the assumptions more explicitly as one of the limitations, and discuss the (positive) societal impacts in the revised version.

---

> > ### Comment · Reviewer_9MCa · 2024-08-13
> >
> > After reading the discussions I'm realizing my reading of thickness and the loss terms was slightly off hence some of my questions were misdirected. First let me make sure I'm viewing this right now: is it the case that both top-k thickness and R2ET optimization goal care not about influence within the top k features nor the influence within the bottom non-top-k features? In this case, your last comment that there might be "mismatch between R2ET’s optimization goal and thickness" is confusing. Was that point about thickness other than top-k ranking thickness of Definition 4.2? 4.2 thickness seem to have a direct representation in the loss term, the one with \lambda_1 coeff of Equation (6).
> >
> > Also; I have now some deeper concerns about the importance of setting k in the methodology. The implicit goal when using some value of k is that we want a model that has k important features and n-k non-important features for each explanation. Given that explanation faithfulness is among the goals of explanations in general and in this paper, the setting of k is as much, if not more, about the problem being solved/predicted by the model than it is about the complexity/interpretability of explanations. If k is too small or too large, the regularization will force the model to do some weird things. Some tasks might not even have a fixed k number of inputs that are important uniformly (i.e. 1 feature is necessary to make prediction for some set of instances while 2 features are necessary for a different set). Can you comment on how to deal with this aspect of the methodology?

---

> > > ### Author Response · Authors · 2024-08-13
> > >
> > > Thanks for the further discussion.
> > >
> > > > is it the case that both top-k thickness and R2ET optimization goal care not about influence within the top k features nor the influence within the bottom non-top-k features?
> > >
> > > We motivate the research based on the P@k (one of the widely used ranking-based metrics), which does *not* care about the rankings within the top-k features, nor the rankings within the remains. Thus, we naturally do not intend to consider such relative rankings for top-k thickness and R2ET.
> > >
> > > > In this case, your last comment that there might be "mismatch between R2ET’s optimization goal and thickness" is confusing. Was that point about thickness other than top-k ranking thickness of Definition 4.2? 4.2 thickness seem to have a direct representation in the loss term, the one with \lambda_1 coeff of Equation (6).
> > >
> > > R2ET's objective in Eq. (6) and thickness in Eq. (3) in Definition 4.2 are not exactly the same.
> > >
> > > Let us go back to the *pairwise* thickness defined in Eq. (2) in Definition 4.1: The *pairwise* thickness measures the explanation robustness by comprehensively capturing the surrounding gaps (by expectation and integration).
> > >
> > > The top-k thickness, defined by pairwise thickness, accumulates the pairwise thickness of all possible top-bottom pairs. Thus, an **exact** computation of top-k thickness will involve expectation and integration due to Eq. (2).
> > >
> > > R2ET optimizes the *bounds* of the top-k thickness based on Eq. (5), rather than the definition of the top-k thickness directly.
> > > $h(x,i,j)$ in Eq. (6) refers to the *gap*, rather than *thickness*.
> > > Thus, optimizing R2ET's objective function does not essentially lead to the best thickness.
> > >
> > > > I have now some deeper concerns about the importance of setting k in the methodology. The implicit goal when using some value of k is that we want a model that has k important features and n-k non-important features for each explanation. Given that explanation faithfulness is among the goals of explanations in general and in this paper, the setting of k is as much, if not more, about the problem being solved/predicted by the model than it is about the complexity/interpretability of explanations. If k is too small or too large, the regularization will force the model to do some weird things. Some tasks might not even have a fixed k number of inputs that are important uniformly (i.e. 1 feature is necessary to make prediction for some set of instances while 2 features are necessary for a different set). Can you comment on how to deal with this aspect of the methodology?
> > >
> > > It is a great question. We believe it is asking for a more "fine-grained" explanation robustness. In a nutshell, we can generalize R2ET to a more "fine-grained" version, although it may go far away from some practical scenarios (e.g., credit card applications only disclose a fixed number of reasons for final decisions).
> > >
> > > As discussed in Lines 107-110, we can generalize R2ET (based on precision@k) to address broader requirements and/or properties. For example, the average precision@k (AP@k) or discounted cumulative gain (DCG) takes the relative rankings/positions among top-k (or the bottom) features into account. To incorporate these metrics, R2ET can be easily extended by setting different weights to various feature pair rankings. Intuitively, the key difference between R2ET (P@k) and the adaptations (based on AP@k and DCG) will be the weights of gaps $h(x,i,j)$ in Eq.(6), and R2ET adaptations encourage to maintain rankings between *any* feature pairs.
> > >
> > > In the case of AP@k or DCG, setting $k=n$ ($n$ be the number of features) is meaningful, and R2ET models will keep reasonable gaps between *any* feature pairs. In other words, models can provide the ranking of every feature's importance.
> > >
> > >
> > >
> > > Thanks again for your time and consideration. Please let us know if further concerns.

---

### Official Review · Reviewer_X6uh · 2024-07-23

**Soundness:** 2
**Presentation:** 3
**Contribution:** 3
**Rating:** 6
**Confidence:** 2

**Summary:**

In this work, the authors study explanation robustness particularly for saliency-based explanations based on gradient information. They propose to use a robustness metric based on the saliency ranking of features. The central benefits claimed (taken from the introduction_ for this approach are:
* Relying on $\ell_p$ metrics is not a good/reliable proxy for robustness
* Attacking $\ell_p$ metrics leads to an arms race between attackers and defenders
In their methodology they provide an alternative called R2ET, the ranking based robustness metric, and describe how this is linked to certified prediction robustnes and adversarial training. Moreover, they state the optimization problems one must solve to compute the R2ET metric.
Experimentally, the authors explore tabular and image datasets and across various models compute the described R2ET metric.

**Strengths:**

The paper is well written in that the authors make their motivations and methodology clear. They also attack an interesting and worthwhile problem in explanation robustness and have put substantial time into their experiments which leads to a comprehensive evaluation of the proposed metric.

**Weaknesses:**

The two critical weaknesses I see with this work are its motivation and evaluation.

Motivation:
Point 1) The authors first motivation is that explanations within a small $\ell_p$ norm can flip all of the rankings and have significantly different "top" features, thus we should use a ranking approach. However, this exhibits the fallacy of "begging the question." Explanations within a small $\ell_p$ norm ball can only appear different when visualized with a heatmap if the values in each feature dimension are very small.^1 This is because what the heatmap shows us is a "ranking" of the highest magnitude features, thus the argument boils down to: "Because small $\ell_p$ norm explanations have a non-robust ranking, we should look at a robust ranking approach" which seems like a poor motivation given that explanations with a small $\ell_p$ norm are not the only kinds of explanations that exist unless your classifier is extremely flat.

Point 2) The authors state that attacking $\ell_p$ robustness leads to an arms race, but also cite [89] which is a formal certification method that proves no attacker exists and therefore thwarts the arms race (although only for small models). If the authors wish to claim the attackers arms race as a motivation for this paper I think they need to do more to (1) show that an arms race will not exist for their method which is not obvious to me and (2) need to show that they scale far beyond [89] which it appears they already do, but this point will need to be made explicit

Point 3) I think the authors should introduce some toy experiments to show where and how their approach really benefits compared to gradient-based explanations. My intuition is that the only benefit of this approach is when the magnitude of the input gradient is small. Is this the authors intuition as well?

^1 I suspect if the authors add values to the colormap under the right-most portion of figure 1 we will see this. Because the

**Questions:**

It is not impossible that my main three points in the weaknesses section stem from a misunderstanding. Do the authors think that I have missed something or misrepresented any of their points?

**Limitations:**

The primary limitation of the paper is in its motivation.

---

> ### Author Rebuttal · Authors · 2024-08-07
>
> > To Weakness 1: The authors first motivation is that explanations within a small L-p norm can flip all of the rankings and have significantly different "top" features ...
>
> 1. **There is a misunderstanding of what the L-p norm is measuring** and let us clarify that. The “small L-p norm” is **not used to measure the magnitude of the explanation itself**, but to **qualify the distance between an original and its perturbed counterpart**. Our study is driven by the observations (Fig. 1) that a “slightly manipulated” explanation (with *a small L-p distance to the original explanation*) can lead to *extremely different feature rankings* from the original ranking, altering human perception and decision-making processes. Thus, we study ranking-based metrics aligning with human perception of salient features.
>
> 2. The above response is **irrelevant** to the magnitude of the original explanation (e.g., no matter whether the gradient’s L-p norm is large or small). As detailed in Proposition 4.5, changes in rankings depend on the joint effect of the *differences in gradient values* and the *Hessian matrix*, rather than the value of the L-p norm.
>
> 3. In our experiments, the classifiers are not necessary to be (extremely) flat since we impose no such constraints and assumptions on the explanations, and R2ET has shown to be effective generally.
>
> > To Weakness 2: The authors state that attacking L-p robustness leads to an arms race...
>
> 1. Theoretically, Proposition 4.5 delineates that the defender, e.g., R2ET, with proper objective function preserves the explanation rankings for **all** types of attackers with a given perturbation budget, and thus mitigates the arms races. Specifically, the proposition shows that the attackers cannot alter the models’ explanation ranking if their perturbation budgets is below the critical threshold (RHS of the inequality). The only chance to manipulate the ranking is to increase the attack budgets. On the other hand, the defenders, e.g., R2ET, increase the critical threshold to prevent **any types of attackers** to mitigate the arms races. In the case of Hessian norm being nearly zero (the denominator goes to zero), it becomes practically impossible for attackers to manipulate rankings successfully. Notice that the proposition is regardless of the “scale” of the models.
>
> 2. Experimentally, Table 1 and Appendix B.2.3 report explanation robustness facing **attackers with different objectives and different attack strategies**, respectively. For all types of attackers, R2ET typically showcases superior robustness under various attacks, indicating the mitigation of the arms races. The datasets and models involved in the experiments are significantly “larger” than those in [89], e.g., the largest feature dimension studied in [89] is **2k** in [89] compared to **396k** in ours.
>
> In sum, we will clarify these statements more explicitly in the revised version.
>
> PS. Our proposed approach focuses on ranking-based metrics for evaluating explanation robustness, and contrasts to the L-p norm-based metrics used in [89].
>
> > To Weakness 3: I think the authors should introduce some toy experiments...
>
> Essentially, R2ET enhances explanation ranking robustness because it maximizes the gaps among features and minimizes Hessian norm, **rather than minimizing the magnitude of the gradient**. Intuitively, wider gaps (larger absolute distances) indicate more significant distinctions in feature importance. The Hessian norm implies the change rate of the feature importance, and a smaller Hessian norm makes it harder for attackers (or any perturbations) to change the model behavior.
>
> Our empirical evidence shows that the key is not the magnitudes of gradients, but their differences. Specifically, in Lines 935-943 (Figures 8 and Table 7) of the appendix, we take the top-2 case of models in Bank and COMPAS in Figure 8 and Table 7 as an example: the model for Bank showcases a smaller average gradient magnitude (approximately 0.08) yet achieves greater top-2 robustness compared to the one for COMPAS, which has a larger gradient magnitude (around 0.3). It indicates that our approach’s benefits extend beyond conditions where gradient magnitudes are minimal.
>
> To address your suggestion regarding toy experiments, which could indeed further validate and clarify the benefits of R2ET. We will consider adding these experiments in the revised version.
>
>
> > To Question: It is not impossible that my main three points in the weaknesses section stem from a misunderstanding. Do the authors think that I have missed something or misrepresented any of their points?
>
> **The most significant misunderstanding is the role of L-p norm. We do not focus on the norm of explanation, e.g., $|I(x)|$, nor small norm explanations. Instead, we investigate whether the L-p distance between the original explanation and its perturbed counterpart, e.g., $|I(x)-I(x’)|$, can serve as a suitable explanation robustness measurement.**
>
> The reviewer seems to misunderstand that we study and restrict the L-p norm of explanation, and doubts the motivation for studying gradient-based explanations with a small L-p norm. However, as Figure 1 shows, our core argument is that even a seemingly minor change (evaluated by L-p norm) in explanations could lead to extremely different explanation rankings (compared to their original explanation ranking). It underscores the necessity for a novel ranking-based approach to assess explanation robustness. **The magnitude of the gradient is not our main focus, and will only marginally impact our theoretical and experimental conclusions.**
>
> ***We would appreciate it if the reviewer could diminish the misunderstanding and re-read the manuscript concerning our response. It would be even better if the reviewer could give more feedback and raise the scores.***

---

> > ### Comment · Reviewer_X6uh · 2024-08-12
> > **Thanks for your comments**
> >
> > I have re-read the paper after the author rebuttal and several of my concerns have been addressed. I still have a few minor concerns, but given the review period is coming to a close I will give the authors the benefit of the doubt and raise my score to reflect my overall postive impression of the paper.

---

> > > ### Author Response · Authors · 2024-08-12
> > >
> > > Thank you for taking the time to revisit our paper and for further considering our manuscript. We greatly appreciate your positive reassessment and for raising the score based on the favorable impression of our work.

---

### Decision · Program_Chairs · 2024-09-25

**Decision:**

Accept (poster)

**Comment:**

Reviewers have reached a consensus that the paper has made a good theoretical and experimental contribution to explanation robustness. The paper is well-organized and the proposed perspective of top-k thickness is new. Following reviewers' comments, AC is happy to recommend acceptance for this paper.

However, note that some weaknesses are highlighted by the reviewers. These include:
1. The motivation needs to be further polished.
2. Discuss the computational overhead of the proposed method and its application to more large-scale datasets.

Please take these comments into consideration in the revised version of this paper.